# Four-dimensional trapped ion mobility spectrometry lipidomics for high throughput clinical profiling of human blood samples

Raissa Lerner [1,3], Dhanwin Baker [1,3], Claudia Schwitter[1], Sarah Neuhaus[1], Tony Hauptmann [2], Julia M. Post [1], Stefan Kramer[2] & Laura Bindila [1] ✉

Lipidomics encompassing automated lipid extraction, a four-dimensional (4D) feature selection strategy for confident lipid annotation as well as reproducible and cross-validated quantification can expedite clinical profiling. Here, we determine 4D descriptors (mass to charge, retention time, collision cross section, and fragmentation spectra) of 200 lipid standards and 493 lipids from reference plasma via trapped ion mobility mass spectrometry to enable the implementation of stringent criteria for lipid annotation. We use 4D lipidomics to confidently annotate 370 lipids in reference plasma samples and 364 lipids in serum samples, and reproducibly quantify 359 lipids using level-3 internal standards. We show the utility of our 4D lipidomics workflow for high-throughput applications by reliable profiling of intra-individual lipidome phenotypes in plasma, serum, whole blood, venous and finger-prick dried blood spots.

Lipids play essential roles in many metabolic and physiological processes, as well as in different pathological conditions[1–3]. They serve as a valuable source of candidates for therapeutic agents, diagnostic and predictive disease biomarkers, treatment response evaluation, and follow-up monitoring in clinical research. Despite their pathophysiological relevance, only a few lipids, other than traditional clinical parameters such as cholesterol, triglycerides (TGs), and lipoproteins are used in clinical practice[4]. This trend is thankfully changing due to the rapidly increasing evidence on lipid association with various diseases facilitated by advanced mass spectrometry (MS)-discovery profiling in large sample cohorts[5–8]. A landmark achievement in this direction is the implementation in clinical practice of risk scores using concentrations of circulating ceramides (Cer) and phosphatidylcholines (PC), CERT1 and CERT2 scores, for identifying and stratifying patients at risk of cardiovascular disease and death[8–11].

However, continuous efforts are required to expedite the clinical translation of lipidomic methods, including quantitative assay validation of both targeted and untargeted lipidomic methods, standardization, definition of acceptance criteria for pre-analytical and analytical parameters across methods, cross-laboratory method validations, biomarker validation, and determination of consensus levels of circulatory lipids.

The rapid improvement in MS technology has led to the development of instruments with higher acquisition speeds and mass resolution, e.g., Orbitrap or time-of-flight (TOF), and higher resolution capabilities of molecular ion separation, e.g., ion mobility spectrometry (IMS). Adding a new dimension to the liquid chromatography-(LC) and MS-based molecular separation, IMS separates ions based on their mobility in a carrier buffer gas, rendering an additional separation based on the ions' collisional cross-section (CCS), hence molecular conformation. Inverting the separation principle of IMS led to the

[1]Clinical Lipidomics Unit, Institute of Physiological Chemistry, University Medical Center, Duesbergweg 6, 55128 Mainz, Germany. [2]Data Mining, Institute of Computer Science, Johannes Gutenberg University Mainz, Staudingerweg 9, 55128 Mainz, Germany. [3]These authors contributed equally: Raissa Lerner, Dhanwin Baker. ✉e-mail: bindila@uni-mainz.de

development of trapped ion mobility spectrometry (TIMS), where ions entering the TIMS analyzer are positioned in an electrical field by the drag of a gas flow. By gradually lowering the electrical force, ions elute from the TIMS device according to their ion mobility[12]. With the introduction of the recently developed MS scan mode named parallel accumulation serial fragmentation (PASEF), the speed of analysis and ion usage have been further improved[13]. The PASEF uses TIMS to accumulate precursors according to their mass to charge ratio ($m/z$) and ion mobility before eluting them further into the mass spectrometer. This technique enhances the signal-to-noise ratio and allows simultaneous fragmentation of multiple precursors in a single scan. In combination with LC, TIMS-PASEF enables analysis of the lipidome using 4 different dimensions, i.e., retention time (RT), CCS, $m/z$, and MS/MS spectra, in what is known as 4-Dimensional (4D) lipidomics, at high speed and mass accuracy[14,15].

To advance the clinical translation of 4D lipidomics for prospective (pre)clinical research, however, further developments are necessary to ensure that the 4D lipidomics adheres to the standards of accuracy, precision, robustness, and throughput required for a bioanalytical assay[16,17]. Routine and confident annotation of detected lipid features in high-throughput profiling requires implementation of stringent matching parameters of data quality indicators (RT, accurate mass, CCS, isotopic pattern, and MS/MS spectral libraries), data curation protocols, and extension of 4D lipid databases for structural references, particularly given the increased number of detected lipid features afforded by TIMS-PASEF[14]. Indeed, failure to do so can lead to around 50% false discovery rate (FDR) of lipid annotations[18].

Here, we present a high-throughput 4D TIMS lipidomics platform amenable for clinical profiling of circulatory lipids. Automated lipid extraction is combined with microflow reversed phase ultra-high-pressure LC (UHPLC)-TIMS-MS and PASEF to enable high-throughput. Integrated 4D lipid annotation strategy and TOF MS survey-based quantification using level-3 internal standards allow reproducible lipid profiling in NIST human plasma 1950 and serum 1951c standard reference material (NIST plasma/serum SRM). Following cross-validation by multiple reaction monitoring (MRM) and inter-laboratory comparison, we showcase the applicability of the 4D TIMS lipidomics for reproducible and comprehensive pheno-mapping in clinical samples by a pilot study of intra-individual and multidien lipidome in blood matrices: plasma, serum, direct blood, fingertip DBS (dried blood spot), and venous DBS.

## Results

### Automatic high-throughput extraction platform and high-throughput LC-TIMS profiling

To develop a workflow for automatic high throughput lipid extraction combined with high confidence lipid identification and quantification, we first adapted the liquid–liquid extraction (LLE)-based method for plasma lipids on a robotic platform. NIST plasma and serum SRM[19] were used for workflow development in order to reference lipid annotation and quantified values to the ones reported by inter-laboratory studies[20,21]. Methyl tert-butyl ether (MTBE)-based extraction was chosen since it has been demonstrated to render comparable lipid recoveries[22,23], and due to its ability to build an upper non-polar lipid phase, which is especially advantageous for fast and clean removal of the organic solvent using a robotic device. One of the critical steps during lipid extraction using a robotic device is the organic phase removal. Optimization of the robotic handling station parameters, in terms of pipetting volume, depth, and speed of pipette immersion in the organic phase allows maximal removal of the organic solvent without contamination with the polar phase.

The recovery and matrix effects of representative internal standards (ISTDs) for different phospholipid (PL), sphingolipid, and glyceride classes obtained using MTBE-based automated extraction were found to be comparable to the ones obtained by manual extraction

(Table 1) and previously obtained using mouse plasma[22], with coefficient of variation (CV) values for relative lipid levels comparable between the two methods (Fig. 1a).

Normalized peak areas of representative lipids to ISTDs show good reproducibility over 32 plasma samples extracted with the robotic platform, with the majority exhibiting CV values below 10%. Only Cholesterol and fatty acyl esters of fatty acid (FAHFA) were found to have CV values above 15% (16.0%; 17.7%), while Cer d18:1_16:0, hexosyl ceramide (HexCer) d18:1_16:0, and phosphatidylserine (PS) 18:0_20:4 range between 11.4 and 12.1% (Fig. 1b), indicating also a good stability of the samples throughout the automatic extraction (Supplementary Fig. 1). Decreased processing time per sample batch and controlled temperature in the handling station likely contribute to the sample stability. The overall processing time was reduced to 3 h per 96-well plate, subsequently reducing the time window variability for (pre)processing and extraction between samples. The fact that the CVs of normalized lipid peak areas lie well under 20% for all assessed lipid classes, thus falling within the accepted CV criterion for bioanalytical assays[16,17], evidence the suitability of the high-throughput lipid extraction and UHPLC/electrospray ionization (ESI)-TIMS-MS method for large-scale analyses. Accordingly, the automatic lipid extraction protocol provides the throughput necessary for large-scale studies with low variability of lipid levels and processing time across multiple samples, thus increasing the standardization level of the lipid extraction (Fig. 1b).

High ionization efficiency, with reduced adduct formation in both ionization modes, and efficient chromatographic resolution within 20 minutes was obtained using micro-flow UHPLC/ESI-TIMS-MS. In fact, chromatographic and ion mobility resolution (Fig. 2a) of lipids exhibit a similar separation pattern as reported with nano-LC-TIMS[14] (Supplementary Fig. 2) and, additionally, allows marginal TIMS-based separation of isomers such as cis/trans (Fig. 2b); thus, enabling a deep annotation of structural diversity of the plasma lipidome from high-throughput profiles.

Table 2 shows the inter-assay reproducibility of the 4D feature detection from repetition experiments performed using automatic lipid extraction and 4D LC-TIMS analysis, corresponding to the duplicate analysis of 32 NIST plasma SRM samples. The overall precursor mass accuracy of all detected lipids was highly accurate with a median value of 0.58 parts per million (ppm) for inter-day analyses and average values per lipid class spanning from 0.25 and 0.99 ppm, respectively. The average CV values for RTs ranged from 0.12 to 0.33% and the average $^{TIMS}$CCS values from 0.09 to 0.18% for the inter-day remeasurement. Altogether, the established workflow shows a high inter-day reproducibility with median variability of 0.58 ppm, and median CVs of 0.19% RT and 0.11% CCS. The inter-day variability of CCS (average CV less than 0.18%, Table 2) compares well with the one reported by Vasilopoulou et al.[14], demonstrating once again the instrument-independent precision of CCS determination, hence enabling cross-laboratory reference of CCS-lipid values and inferring CCS-based lipid annotations for TIMS data irrespective of the chromatographic method used (see nano LC-TIMS-MS and UHPLC-TIMS-MS). Although reversed-phase chromatographic resolution is superior to mobility resolution (Fig. 2b), the TIMS-based separation could allow further reduction of chromatographic separation time while preserving annotation coverage using CCS values. Also, TIMS- separation could prospectively aid in annotation coverage of isomeric species when coupled to other chromatographic media such as HILIC separation or to chip-based infusion.

The stability of the plasma lipids elution profiles and good inter-day reproducibility of the 4D feature detection (Fig. 1b and Table 2) permitted the implementation of stringent variability criteria for RT-based lipid annotation and curation, which, in addition to accurate RT values obtained for the 200 lipid standards, substantially reduced the number of false-positive structural assignments[24]. Similarly, the low variability of CCS values permitted inclusion of the CCS deviation as an additional data attribute for confident automatic annotation (Fig. 3).

**Table 1 | Recovery and matrix effects for NIST plasma SRM automated extraction vs manual**

| Analyte | Automated extraction | | Manual extraction | | No extraction | |
|---|---|---|---|---|---|---|
| | Recovery % | Matrix effect % | Recovery % | Matrix effect % | Recovery % | Matrix effect % |
| Cer d18:1_16:0 d7 | 80.5 | 82.0 | 59.1 | 57.0 | 68.7 | 39.2 |
| Cer d18:1_17:0 | 87.1 | 62.5 | 69.6 | 54.4 | 72.9 | 39.7 |
| Cer d18:1_18:0 d7 | 86.1 | 40.0 | 57.1 | 67.2 | 60.1 | 40.4 |
| Cer d18:1_24:0 d7 | 110.5 | 111.3 | 71.2 | 102.4 | 9.0 | 9.5 |
| 5-PAHSA d9 | 81.4 | 84.4 | 67.6 | 83.1 | 65.0 | 54.1 |
| 9-PAHSA d9 | 84.3 | 98.2 | 73.7 | 99.0 | 78.6 | 77.9 |
| LPC 17:0 SN1 d5 | 51.0 | 95.6 | 46.7 | 99.2 | 67.8 | 67.3 |
| LPC 17:1 SN1 | 37.7 | 88.0 | 35.9 | 112.3 | 57.6 | 64.7 |
| LPE 17:0 SN1 d5 | 74.3 | 98.5 | 67.1 | 104.5 | 76.9 | 80.4 |
| LPE 17:1 SN1 | 61.7 | 96.4 | 62.1 | 92.6 | 78.3 | 72.4 |
| PC 17:0_20:3 d5 | 82.8 | 64.8 | 78.7 | 79.6 | 75.7 | 60.2 |
| PC 17:0_14:1 | 86.2 | 70.9 | 76.2 | 65.4 | 73.1 | 47.8 |
| PC 17:0_14:1 d5 | 82.6 | 51.3 | 76.0 | 55.6 | 75.8 | 42.2 |
| PC 17:0_18:1 d5 | 81.1 | 70.2 | 76.4 | 65.3 | 79.8 | 52.1 |
| PE 17:0_14:1 | 81.5 | 110.6 | 80.7 | 66.9 | 82.5 | 55.2 |
| PE 17:0_14:1 d5 | 80.7 | 107.9 | 75.9 | 66.5 | 80.6 | 53.6 |
| PE 17:0_18:1 d5 | 83.3 | 57.9 | 67.7 | 63.7 | 71.8 | 45.7 |
| PG 17:0_18:1 d5 | 80.0 | 32.9 | 66.5 | 36.1 | 84.0 | 30.3 |
| PG 17:0_14:1 | 76.1 | 84.5 | 73.4 | 98.9 | 82.9 | 82.1 |
| PI 17:0_14:1 | 68.4 | 94.4 | 59.5 | 101.9 | 87.9 | 89.6 |
| PI 17:0_14:1 d5 | 68.7 | 96.0 | 56.1 | 106.5 | 86.1 | 91.6 |
| PI 17:0_18:1 d5 | 73.1 | 53.5 | 65.5 | 89.9 | 83.2 | 74.8 |
| PI 17:0_22:4 d5 | 70.5 | 39.4 | 69.8 | 63.3 | 91.9 | 58.1 |
| PS 17:0_14:1 | 75.2 | 72.8 | 72.9 | 108.6 | 98.3 | 106.8 |
| PS 17:0_14:1 d5 | 72.9 | 78.1 | 68.2 | 110.8 | 94.1 | 104.3 |
| PS 17:0_18:1 d5 | 81.5 | 40.2 | 67.3 | 41.2 | 108.6 | 44.7 |
| SM d18:1_24:1 d9 | 85.3 | 69.7 | 74.4 | 70.8 | 77.5 | 54.9 |
| SM d18:1_12:0 | 72.4 | 52.1 | 70.9 | 96.7 | 74.5 | 71.9 |
| SM d18:1_16:1 d9 | 66.3 | 62.7 | 70.2 | 89.1 | 75.9 | 67.7 |
| SM d18:1_18:1 d9 | 80.5 | 83.0 | 78.1 | 89.4 | 80.1 | 71.6 |
| Cholesterol d7 | 82.6 | 59.5 | 84.4 | 31.7 | 82.8 | 26.2 |
| TG 14:0_16:1_14:0 d5 | 103.4 | 48.3 | 97.9 | 65.9 | 83.8 | 55.2 |
| TG 17:0_17:1_17:0 d5 | 89.8 | 36.8 | 87.6 | 29.4 | 77.8 | 22.9 |
| TG 19:0_12:0_19:0 d5 | 126.5 | 75.6 | 145.3 | 40.3 | 136.8 | 55.1 |

Comparison of recovery and matrix effects for automated, manual, and no-extraction of NIST human plasma standard reference material (SRM). For estimation of the recovery, NIST plasma SRM samples were spiked with set 2 deuterated internal standards (ISTDs) pre- ($n = 64$) and post- ($n = 8$) automated and manual extraction ($n = 8$). For the No-extraction, 20 µl of plasma were diluted 1:20 and spiked to the same final concentration. Recovery was calculated in percentage using the averages of ISTD peak areas from the samples spiked prior to and without extraction to the ISTD peak areas from the samples spiked after extraction, respectively. For matrix effect evaluation, ISTD peak areas of set 2 spike solution for automated ($n = 21$) and manual ($n = 8$) workflow were compared to ISTD peak areas in NIST plasma SRM samples spiked post and without extraction. Source data are provided as a Source data file.

The high-throughput extraction method combined with the UHPLC-PASEF-MS reported here allows highly reproducible profiling of plasma or serum lipid extracts. The overall time for sample (pre)processing, extraction, and analysis, including preparation and analysis of calibration curves, quality control (QC), analysis and wash runs, as well as data processing, is estimated at less than 3 days for 96 samples.

**In-house library and lipid identification**

Many lipid databases are available to identify and annotate lipid species; however, a significant number lack mobility information. The freely available web server-based LipidIMMS analyzer[25] is an important annotation tool for IM-MS-based lipidomics, which includes also RT calibration to improve the confidence of lipid annotation. The CCS predict tool[26] imbibed within the LipidIMMS analyzer predicts CCS values that usually exceed our CCS accuracy and tolerance levels for identification (Supplementary Table 1, Fig. 3). For instance, for PC 16:0_18:1 a CCS value of 292.5 Å$^2$ was obtained by our analysis whereas the predicted value is 286.4 Å$^2$ (Supplementary Table 1). Within this CCS interval, we detected 50 features in plasma out of which 39 were annotated. Therefore, CCS predict was exclusively used to infer CCS differences between lipid isomers as a means to verify annotation.

Hence, the in-house library was generated and a feature selection strategy was devised to achieve confident and routine annotation; i.e.,

reduce false-positive assignments and data curation time, thus streamlining the structural assignment of the TIMS lipid data. The in-house library consists of 150 lipid standards in positive mode and 158 lipid standards in negative mode, covering altogether 16 major lipid classes (Fig. 3). In negative mode, [M-H]$^-$ was the prominent ion for all lipid classes except for lyso phosphatidylcholine (LPC), PC, sphingomyelin (SM) for which [M + HCOO]$^-$ is formed due to the presence of quaternary amine group in their structure. In-source demethylation of PC species does not occur, as evidenced by extracted ion chromatograms of PC 34:2 as a formate adduct (at $m/z$ 802.5604) and its demethylated counterpart (at $m/z$ 742.5392) (Supplementary Fig. 3). [M + H]$^+$ was the prominent ion in positive mode for all lipid classes, except for phosphatidylglycerol (PG), cholesteryl ester (CE) and TAG for which [M + NH$_4$]$^+$ is formed. [M-H$_2$O + H]$^+$ was observed as the second most abundant ion for Cer species only (Supplementary Fig. 4), whereas [M + Na]$^+$ and [M + K]$^+$ ions were observed at low abundance for most of the lipid classes. The provision of accurate measurements of CCS, RT, and $m/z$ values for individual lipid standards in both ionisation modes expedited curation and creation of a reference 4D plasma library for high throughput sample annotation. For example, accurate RT, CCS, and $m/z$ values for individual PC standards in positive ion mode helps in delineating particular PC isomers of a given PC composition from multiple possible ones with identical fragmentation

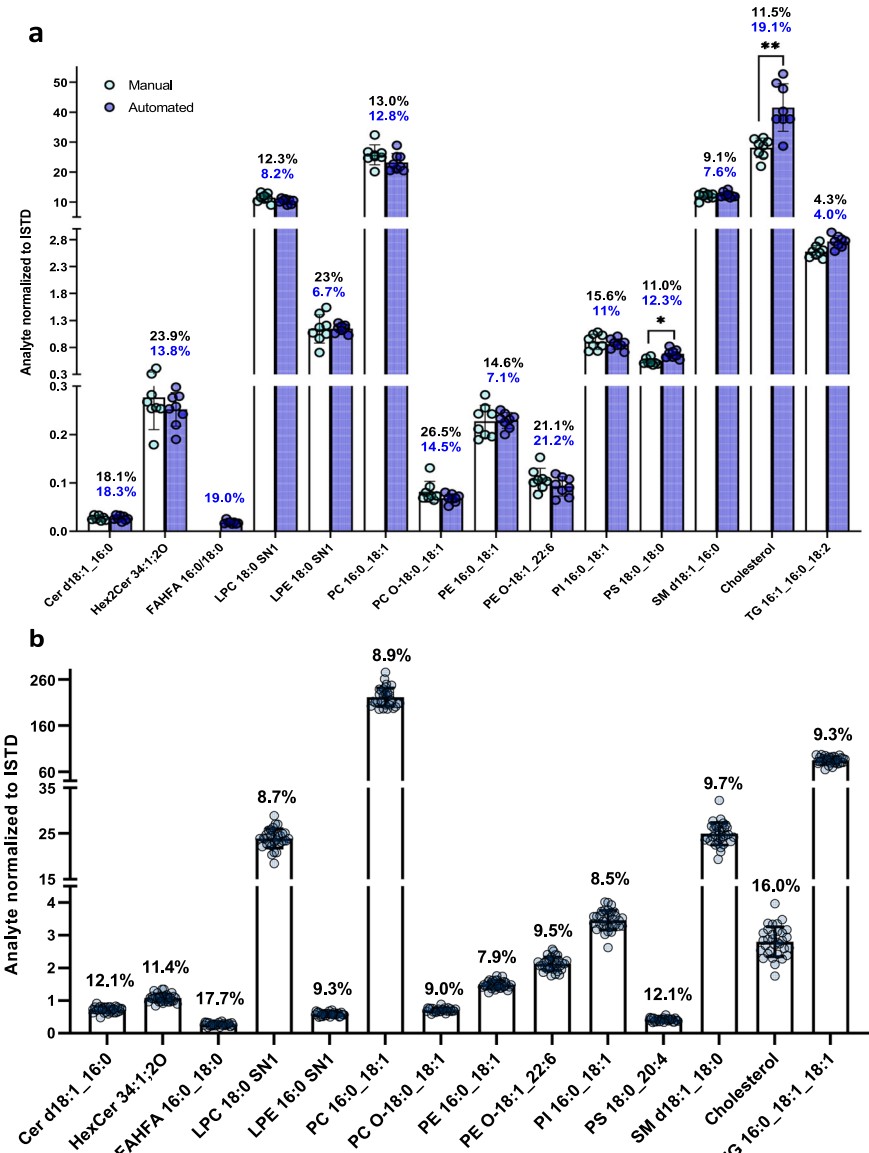

**Fig. 1 | Comparison of automated vs manual extraction of NIST human plasma standard reference material (SRM) and assessment of analysis reproducibility. a** Direct comparison of automated vs manual extraction. Depicted are analyte peak areas, normalized to respective internal standard (ISTD) peak areas, of representative lipids for each analyzed lipid class, whose extraction was either manual (light blue dots, empty bars) or automated (dark blue dots, blue bars) from NIST plasma SRM. Each dot represents an independent extraction ($n = 8$). Black numbers indicate the coefficient of variation (CV) values for manual extraction, whereas blue values indicate CV for automated extraction. Two-tailed multiple unpaired t-test with multiple comparison correction (Holm−Sidak method) was used ($t_{PS\ 36:0}(14) = 4.111$, $p = 0.012630$; $t_{Cholesterol}(14) = 4.424$, $p = 0.007482$). Bars represent mean values ± standard deviation (SD). *$p < 0.05$; **$p < 0.01$. **b** Depicted are values from representative lipid species extracted from NIST plasma SRM, normalized to their corresponding ISTDs. Measurement was conducted in negative ion mode. The displayed numbers show the CV. Each dot represents an independent extraction ($n = 32$). Bars represent mean values ± (SD). Source data are provided as a Source data file and in Supplementary Data 6.

patterns. The 4D descriptors for lipid standards were found to be instrumental to infer RT and CCS differences between lipid isomers and isobars as a reference to confirm and/or expand lipid annotation in plasma (Supplementary Data 1). Lack of highly confident annotations of lipid structures using both RT and CCS values can readily lead to misannotations in complex matrices. Recent lipidomic publications[14,18,27] also underlined that the number of annotation entries for a given lipid composition often exceeds the number of biologically possible structures and that inclusion of accurate RT values for lipid annotation as applied in this study can alleviate such pitfalls. Similarly, CCS values were valorized here as additional quality data attributes for confident lipid annotation (Fig. 3) and exclusion of misannotations due to tailing or multiple hits (see Supplementary

Data 2 for examples of CCS-based curated annotations). Such issues were addressed using the in-house library of lipid standards and plasma lipids in conjunction with feature selection strategy (Fig. 3).

By the addition of manually curated lipid species from the NIST plasma SRM, we expanded the library to 391 lipids in negative ion mode and 424 lipids in positive ion mode, accounting for a total of 801 entries (Supplementary Data 3). For this curation, rule-based lipid fragmentation, spectral matching to MS Dial library, and RT- and CCS-differences between lipid species inferred from the lipid standards library and CCS predict tool were used (Fig. 3).

The expanded library with all the 4D descriptors (RT, *m/z*, CCS, fragmentation spectrum), named ClinLip Analyte List (Supplementary Data 1 and 3) from hereon, enables fast lipid assignment, e.g.,

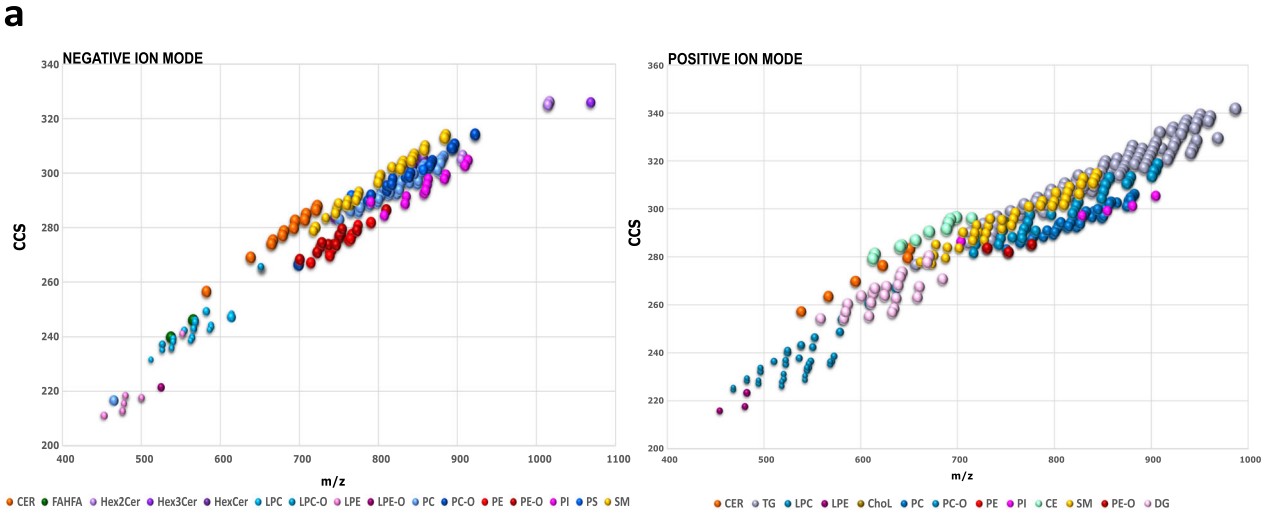

**a**

**b**

**Fig. 2 | Comparison of chromatographic vs mobility separation. a** Three-dimensional (*m/z*, collisional cross section (CCS), retention time (RT) (bubble size) distribution of identified lipids from NIST human plasma standard reference material (SRM) in negative (left) and positive (right) ion mode (average value of each dimension from *n* = 32 sample measurements in each mode).

**b** Chromatogram and mobilogram traces for the separation of a 1:1 synthetic mixture of cis and trans phosphatidylcholine (PC) PC 18:1_18:1 and of cis and trans phosphatidylglycerol (PG) PG 18:1_18:1, run over the 20 min profiling method in neg ion mode. Source data are provided as a Source data file.

**Table 2 | Inter-assay reproducibility NIST plasma SRM automated extraction**

| Analyte | Mass error ppm | RT CV % | CCS CV % |
|---|---|---|---|
| Cer | 0.70 | 0.12 | 0.10 |
| HexCer | 0.99 | 0.12 | 0.09 |
| FAHFA | 0.29 | 0.30 | 0.18 |
| PC (-O) | 0.89 | 0.14 | 0.10 |
| SM | 0.68 | 0.15 | 0.09 |
| PS | 0.44 | 0.15 | 0.11 |
| PE (-O) | 0.25 | 0.12 | 0.10 |
| PI | 0.59 | 0.14 | 0.09 |
| LPC (-O) | 0.42 | 0.33 | 0.11 |
| LPE (-O) | 0.59 | 0.28 | 0.11 |

Inter-assay reproducibility. Reproducibility of mass accuracy [m/z], retention time (RT) [min], and collisional cross section (CCS) [Å²] was assessed by remeasuring the NIST human plasma standard reference material (SRM) in negative ion mode after 2 weeks and by evaluating all identified lipids from both experiments (n = 64). The reproducibility of CCS and RT was calculated as the coefficient of variation (CV), whereas the mass error is given in parts per million (ppm). Displayed values are the average CVs for all analyzed lipids within one lipid class. Source data are provided in Supplementary Data 6.

within a few minutes, with high confidence score for each of the data attributes embedded in the annotation score: precursor mass deviation, RT deviation, isotopic pattern quality, MS$^2$ match, and CCS deviation.

Compared to the MS Dial spectral library, the ClinLip Analyte List renders increased lipid annotation accuracy as exemplified for phosphatidylethanolamine (PE) species. Use of only m/z- and spectral matching-based annotation, as currently afforded by MS Dial embedded in the data processing software, led to false-positive identification of several features as lyso N-acylphosphatidylethanolamines (LNAPE) instead of PE (Supplementary Table 2). The benefit of including RT as an annotation criterion is obvious for isobaric/isomeric structures such as lyso sn1/sn2 isobars with fragmentation patterns indistinguishable by spectral matching. RT differences between lyso sn1/sn2 isobars inferred from literature[28,29] allowed their distinct annotation across lyso-species (Supplementary Data 1 and 3 and Supplementary Fig. 5). CCS values differ less than 1 A$^{o2}$ for the distinct sn1/sn2 lyso isomers and hence, are not baseline or partially separated by mobility. Noteworthy, extracted ion mobilograms (EIM) for lyso species resulted in two overlapping but distinct mobilograms and areas thereof, closely resembling the chromatographic peak area differences. Thus, future developments in software-assisted analysis of overlapping mobilograms in TIMS data could enhance the annotation coverage of low abundant lipids[30] irrespective of the chromatographic resolution in untargeted lipidomics.

Although MS Dial provides a very comprehensive lipid library, valorized here for initial curation of plasma lipid annotation, the ClinLip Analyte List combined with feature selection strategy (Fig. 3) substantially reduced the number of hits and false annotations and allows reliable and comprehensive lipidome coverage. False discovery rate depends highly on the number of samples per annotation bucket, the ionization mode, and the feature selection parameters; particularly with the filters for feature recursiveness across sample batch and ranges for mass accuracy, CCS and RT deviation, and MS/MS matching score (Fig. 3). For example, using the stringent feature selection parameters detailed in (Fig. 3) in conjunction with ClinLip Analyte list for n = 32 extracts, a 2.5% of lipids in negative ion mode were manually corrected and 5% additional hits representing misannotations or entries without reference RT and CCS values were discarded. Use of CCS but not RT as an annotation parameter rendered a 10% increase in hits number requiring subsequent curation, emphasizing the utility of 4D descriptors for automatic lipid annotation.

As a result, we were able to annotate with high confidence (in n = 32 NIST plasma SRM extracts) 370 lipids from which 55 were uniquely found in negative and 191 uniquely found in positive ion mode (Supplementary Fig. 6). Out of the 370 lipids annotated in plasma, 359 were later qualified for quantification. 364 lipids were also identified with the ClinLip Analyte List in the NIST serum SRM without any further adaption of the annotation scores or threshold filters, amounting to over 90% of the identified lipids in the NIST plasma SRM (Supplementary Data 4). Prospective improvement in 4D feature extraction algorithm, annotation software, and expansion of 4D library are envisaged to further increase coverage and confidence in automatic lipidome annotations in routine profiling.

Resolution of isomers by reversed-phase chromatographic separation, even for a 20 min separation time, is superior to TIMS separation (Fig. 2b). The 4D data processing software was found to have certain limitations on the delineation of partially separated or co-eluting isomers. In synthetic mixture, TG 16:0_18:1_20:4 and TG 18:1_18:2_18:2 were partially chromatographically separated, with distinct PASEF spectra, but not separated in the broad ion mobility resolution survey. Unfortunately, TG 16:0_18:1_20:4 and TG 18:1_18:2_18:2 isomers were not extracted as distinct 4D features by the data processing software neither in synthetic mixture nor in plasma and hence, their annotation required inspection and delineation from the raw data. In ultra-ion mobility resolution mode these isomers can be delineated including by distinct PASEF spectra over the two mobilogram peaks (Supplementary Fig. 7), showing that targeted ion mobility separation in ultra-ion mobility resolution mode can be valorized for isomer annotation. Similar results were obtained for these isomers in plasma (Supplementary Fig. 7). The isomers TG 18:1_18:1_20:4 and TG 18:1_20:4_18:1 in synthetic mixture or in plasma were not chromatographically and/or ion mobility-based resolved (Supplementary Fig. 8). From the isobaric TG 18:2_20:4_22:6 and TG 20:4_20:4_20:4 species, only TG 18:2_20:4_22:6 was unambiguously delineated in plasma (Supplementary Fig. 9).

Finally, out of 15899 features initially detected from NIST plasma SRM in negative ion mode, 470 highly reproducible stable unknown features, exhibiting a plasma dilution response, were derived (Fig. 3 and Supplementary Data 5). Structural references for these species were not found even when using MetaboBASE Personal Library 3.0. Nevertheless, accurate selection of the stable unknown features enables prioritization of targets for additional structural elucidation and are, therefore, highly relevant for prospective analyses to increase the annotation coverage of the stable lipidome.

## Molar quantification and ISTDs performance

Each lipid class exhibited at least 64 folds difference between the lowest and the highest point in the linear concentration range (Fig. 4a). This dynamic range, in addition to the good sensitivity[22] (Fig. 4a) and reproducibility of detection (Fig. 1b), indicated the amenability of multiple lipid classes for quantification in complex biological matrices.

A variety of MS-based methods using reverse or hydrophilic LC-MS[31–34] or chip-based shot-gun MS approaches[16,35–37] have been used to quantitate plasma lipidome. The different approaches result in variable lipidome coverage, specificity, and sensitivity of the analysis[38]. The lipidomic community reached a consensus opinion whereby the choice and performance of ISTDs[39–42] and computational tools[43] highly influences the lipid quantification output. The use of one or more ISTD per individual lipid species[44], such as Labelled Isotope Labelling of Yeast (LILY) mixture[45] consisting of an isotopically labeled yeast lipidome extract, enabling the use of so-called Level-1 ISTDs for every investigated analyte was proposed[33], but the strategy is restricted to yeast extracts and its use in mammalian plasma was not evidenced. The increased matrix complexity rendered by such a mixture challenges efficient ionization and resolution of molecular species, and data processing[46]. We opted for and assessed the use of level-3 class-specific

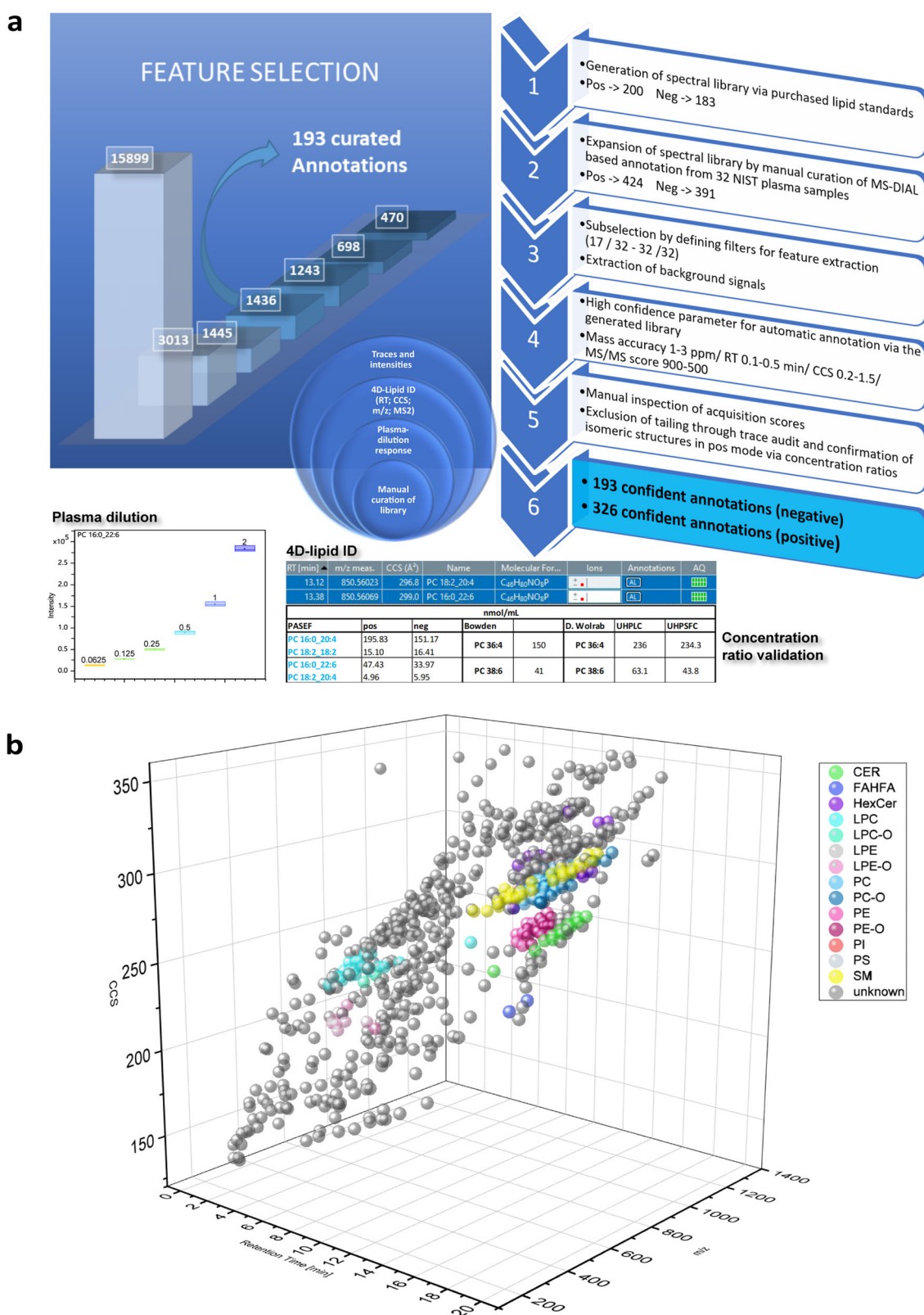

ISTDs (https://lipidomicstandards.org/lipid-species-quantification/)[47] not commonly found in the human plasma (set 1), as well as of deuterated ISTDs (set 2) (Supplementary Tables 3 and 4). The performance of the set 1 ISTDs, previously demonstrated for MRM plasma lipid quantification[22,48], was evaluated here based on the concurrence between MRM and 4D lipidomics platform in both ionization modes and reference to inter- and cross-laboratory studies. The majority of

the set 1 ISTDs performed well using the automatic high-throughput extraction and survey TOF-MS analysis, with few exceptions. For the PE lipid class quantification in positive mode, PC ISTD (Supplementary Table 3) was used because of background interference from an external standard in the calibration mixture leading to a linear response of the PE 17:0_14:1 ISTD. This was observed for different purchased batches of PE 17:0_14:1.

**Fig. 3 | Confident annotation and feature selection. a** Scheme representing the workflow for the manual curation of all the features from a data set (negative ion mode) to the confident annotation of 193 lipid species and the identification of stable unique features. 15,899 features are obtained from the repetition analyses of the 32 NIST human plasma standard reference material (SRM) samples without applying rules for filtering of the features. 3013 features are found in 32 out of 32 analyses when the filter for recursive feature extraction is set to 1 out of 32. 1445 features are found in 32 out of 32 analyses when the filter for recursive feature extraction is set to 17 out of 32. 1436 features are present after background subtraction (methanol/Isopropanol sample). 1243 features are left after the subtraction of 193 annotated features (including internal standards (ISTDs)). 698 features overlap with the dilution experiment, with tolerance values set to <0.002 $m/z$, 0.1

retention time (RT), and 0.2 collisional cross section (CCS). 470 features out of the overlapping features exhibit a dilution response (Pearson correlation ≥0.9 and standard deviation of the mean values from the dilution experiment >0.1). The dilution experiment was performed in triplicates with $n = 3$ measurements per dilution step, ranging from 2–0.0625 µl plasma on the column. The box plots show median and mean values and 25% and 75% quartiles within a colored box. Gray whiskers indicate the lower and upper extremes. **b** The cubic plot shows CCS and RT as a function of $m/z$ for all the features identified in a NIST plasma SRM. All the major identified lipid classes are shown with distinguishing-colored bubbles, whereas the unknown features are marked with gray bubbles. Source data are provided in Supplementary Data 3, 5, 6, and 15.

PC 17:0_14:1 rendered good results in negative mode. In positive ion mode, the quantification using PC 17:0_14:1 can be affected by its natural occurrence in the NIST plasma and serum SRM. Unfortunately, due to the high ion usage and deep structural information obtained via PASEF measurements, other exogenous PC ISTDs recommended for quantification showed endogenous traces in plasma (Supplementary Table 5). The alternative use of PC 15:0_18:1 d7 in positive ion mode leads to overexpressed PC concentrations as also described earlier[20]. PC 17:0_14:1 rendered better values than PC 15:0_18:1 d7 as per interlaboratory and cross-method comparison, where subtraction of endogenous PC 17:0_14:1 did not significantly improve quantification results (Supplementary Table 6). Similarly, endogenous levels of SM d18:1_12:0 were detected in positive mode; however, subtraction of endogenous level was beneficial for quantification (Supplementary Table 3, set 1 ISTD). For SM quantification in negative ion mode, PC 17:0_14:1 was alternatively used to SM d18:1_12:0. The latter when spiked during automatic extraction at a concentration of 75 ng/ml resulted in overestimated SM concentrations, likely attributable to high losses due to a large volume of spiking solution (Supplementary Table 3). The possible isotopic interference of PC 17:0_14:1 with SM 34:1; 3O was excluded due to their difference in retention time (Supplementary Fig. 10). Cholesteryl esters (CE) quantification using TG ISTD rendered reproducible results for $n = 32$ with CV ‹25.4% (Supplementary Data 6 and 7) although, levels differ substantially compared to the inter-laboratory study by Bowden et al.[21] CE peak areas exhibited, similar to TG species, a progressive decay over the multiple extract batch analysis, which was better accounted for by TG than cholesterol ISTD given the stronger hydrophobicity differences between esterified and free cholesterol (Supplementary Fig. 11). Comparison between the quantification outputs using different sets of internal and external standards (set 1 and set 2, Supplementary Tables 3 and 4) leads to conclusions consistent with the lipidomic community's view[39–41] on the influence and choice of ISTDs on the lipid levels. Average CV values ‹25% between the molar concentrations obtained using set 1 and set 2 ISTDs (Supplementary Tables 3 and 4), were obtained for the majority of lipid classes in negative ion mode, except for ceramides, FAHFA, and hexosylceramides, where average CV values (per class) of 31.1%, 64.85%, and 43.35%, respectively, were obtained (Fig. 4b and Supplementary Data 6 and 7). In positive ion mode, cholesterol, LPE, PE, phosphatidylinositol (PI), and SM classes exhibit an average CV ‹25%; Cer and PC exhibit CVs under 35%, while CVs for DG, CE, and TG exceed 30% between the two sets of standards (Supplementary Data 7). These data, along with prospective systematic evaluation of the performance of multiple standards in reference samples of normo- and hyperlipidemia, will guide the choice of ISTDs for large-scale cohort analysis and of the strategy to factor in the ISTDs' performance differences when referencing lipid molar concentrations obtained by 4D lipidomics platform across different assays, sample cohorts, and laboratories. Nevertheless, this evaluation also indicates a good reproducibility of quantification of a sample batch with a given set of ISTDs/calibrants (Fig. 4b, Supplementary Data 6 and 7), and hence the method can be used for lipid marker discovery in clinical samples.

## Multi-point and one-point quantification

To derive the suitability for large-scale analysis, the multi-point and one-point quantification strategies were evaluated on TOF MS raw data from the NIST plasma SRM replicates. The one-point quantification strategy gave rise to highly comparable values to the multipoint strategy for the majority of the lipid classes (CV values of <13%) (Fig. 4c). For LPC and PI species CV values are 21%. For Cer species the one-point calibration rendered larger differences in quantified levels depending on the chosen concentration of the calibrant. This underscores the benefit of multipoint calibration for accurate quantification, where non-linearity of the ion response, particularly at low- and high-end concentration, can be regression-corrected over the dynamic concentration range (Fig. 4c and Supplementary Data 6). Inter-day reproducibility of the multi-point quantification exhibited CV values of <30% (Fig. 4b) for the quantified lipid species over 64 analyses, irrespective of the sets of external standards/ISTDs, with the majority of species exhibiting CV values <20% of molar concentrations (Fig. 4b and Supplementary Data 6 and 7). Inter-day assay with remeasurement of plasma extracts ($n = 32$ extracts and remeasurement thereof after 2 weeks) and inter-plate assay (2x plates, $n = 32$ extracts per plate) exhibited similarly good reproducibility despite one year difference in experiments and the different ISTD sets. Average CV values for TGs shifted from 19% and 27% per 32 extracts to 48.5% for 64 extracts. This shift in CV values of TGs with increasing number of samples and batch analysis time is likely due to their propensity for hydrophobic precipitation with time in LC autosampler; ca 20 h for 32 extracts to ca 48 h for 64 extracts, since PC and PE species do not exhibit similar decay over time. High abundant TGs exhibiting a decay over time were found to be compensated by similarly decaying ISTDs. The majority of low abundant TGs did not show a similar decaying trend, hence are not properly accounted by the decaying ISTD. (Supplementary Fig. 12). Therefore, the design of operational procedures for prospective high-throughput cohort profiling requires consideration of the choice of multiple TG ISTDs, management of residing time of samples in autosampler and temperature conditions in order to mitigate risks for unaccountable hydrophobic precipitation of TGs and CEs (Supplementary Figs. 11 and 12)[49–51].

The good intra- and inter-assay reproducibility per set of level-3 ISTDs/calibrants indicate that lipid quantification using survey TOF MS data and multi-point calibration strategy is reproducible over a broad dynamic range, evidencing the amenability of the 4D lipidomic platform for marker profiling and discovery.

To further benchmark the method's suitability for clinical profiling, we calculated CERT2 scores in NIST plasma SRM using the molar concentrations values of Cer d18:1_24:1/Cer d18:1_24:0, Cer d18:1_16:0/PC 38:5, Cer d18:1_18:0/PC 14:0_22:6, and PC 16:0_16:0 obtained with each of the two standard sets: $n = 32$ plasma extracts with set 1 ISTDs and $n = 64$ extracts with set 2 ISTDs, amounting to a total of 96 NIST plasma SRM extracts processed 1 year apart (Supplementary Data 8). The obtained CERT2 scores are consistent with low risk for each of the two assays of the NIST plasma SRM, supporting the applicability of 4D platform for reliable lipid marker profiling and discovery in clinical samples.

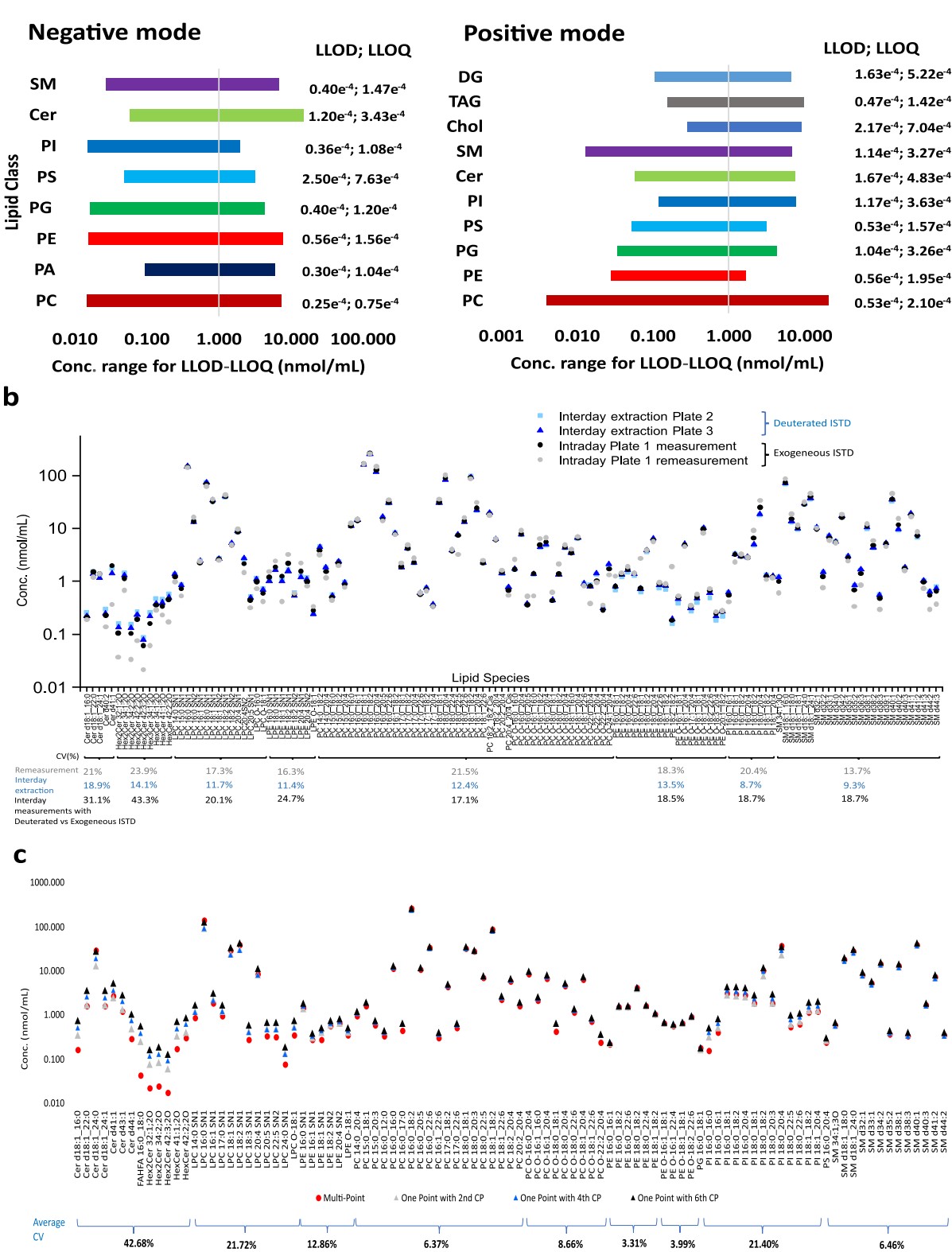

**a**

**Negative mode**

| Lipid Class | LLOD; LLOQ |
|---|---|
| SM | 0.40e⁻⁴; 1.47e⁻⁴ |
| Cer | 1.20e⁻⁴; 3.43e⁻⁴ |
| PI | 0.36e⁻⁴; 1.08e⁻⁴ |
| PS | 2.50e⁻⁴; 7.63e⁻⁴ |
| PG | 0.40e⁻⁴; 1.20e⁻⁴ |
| PE | 0.56e⁻⁴; 1.56e⁻⁴ |
| PA | 0.30e⁻⁴; 1.04e⁻⁴ |
| PC | 0.25e⁻⁴; 0.75e⁻⁴ |

Conc. range for LLOD-LLOQ (nmol/mL)

**Positive mode**

| Lipid Class | LLOD; LLOQ |
|---|---|
| DG | 1.63e⁻⁴; 5.22e⁻⁴ |
| TAG | 0.47e⁻⁴; 1.42e⁻⁴ |
| Chol | 2.17e⁻⁴; 7.04e⁻⁴ |
| SM | 1.14e⁻⁴; 3.27e⁻⁴ |
| Cer | 1.67e⁻⁴; 4.83e⁻⁴ |
| PI | 1.17e⁻⁴; 3.63e⁻⁴ |
| PS | 0.53e⁻⁴; 1.57e⁻⁴ |
| PG | 1.04e⁻⁴; 3.26e⁻⁴ |
| PE | 0.56e⁻⁴; 1.95e⁻⁴ |
| PC | 0.53e⁻⁴; 2.10e⁻⁴ |

Conc. range for LLOD-LLOQ (nmol/mL)

## NIST plasma and serum SRM lipidome quantified by high-throughput 4D lipidomics platform

359 out of 370 lipids annotated with high-confidence in NIST plasma SRM were reproducibly (n = 32 and n = 64) quantified with the 4D lipidomics quantification strategy, i.e., multi-point quantification of TOF MS-survey data using level-3 ISTDs (Supplementary Data 6 and 7). 354 lipids were quantified in NIST serum SRM (Supplementary Data 4).

About 90% of the NIST plasma SRM lipids were also detected in NIST serum SRM, confirming the quantitative and qualitative similarity of plasma and serum reported by other laboratories[20] (Fig. 5a). The good reproducibility of the quantification method is evident for the 160 lipid species commonly identified and quantified in negative ion mode in 32 extracts of each NIST plasma and serum SRM samples (Fig. 5a and Supplementary Data 4).

**Fig. 4 | Bioanalytical method validation and quantification strategies.**
**a** Concentration range and limit of detection (LOD) – limit of quantification (LOQ) of each lipid class determined using representative lipid standards in both ionization modes, that passed the accepted criterion of bioanalytical method validation for the LOD-LOQ determination. **b** Concentration of 125 lipid species in nmol/mL plotted against lipid species identified from the NIST human plasma standard reference material (SRM) samples processed on different days. The black and gray dots represent quantified values, using set 1 standards, from intra-day measurement ($n = 32$ extractions using 1 plate) and remeasurement (of the $n = 32$ extractions), respectively. Light and dark blue dots in the graph represent quantified values, using set 2 standards, from the inter-day extraction analysis ($n = 32$ extractions per plate; 2 plates), respectively. Gray coefficient of variation (CV) values indicate the variation in quantified values for the intra-day measurement

($n = 32$ extractions) combined with the remeasurement thereof ($n = 64$ equivalent to black and gray dots), blue CV values stand for the inter-day extraction ($n = 64$ samples equivalent to light and dark blue dots), and black CV values for the comparison of the first intra-day measurement (set 1 standards) ($n = 32$ samples) with the inter-day extraction (set 2 standards) ($n = 64$ equivalent to black, light and dark blue dots). **c** Concentration of 125 lipid species in nmol/mL plotted against lipid species identified from 32 NIST plasma SRM samples and quantified using multi- and one-point calibration. For the one-point calibration, various calibration points were compared. The red circles represent the quantified values via multi-point quantification whereas the gray, blue, and black triangles represent the quantified values via one-point quantification using 2nd, 4th, and 6th Calibrant point of Set 1 standards. Source data are provided as a Source data file and in Supplementary Data 6 and 7.

In the NIST plasma SRM, 125 species were commonly identified and quantified in both ionisation modes (Fig. 5b and Supplementary Data 6); 55 lipids were uniquely identified and quantified in negative ion mode, and 191 in positive ion mode (Fig. 5c). The concordant quantification results across ionisation modes indicate that this concordance can serve as a valuable internal control of the 4D lipidomics quantification, at least for the 125 common species (Fig. 5b and Supplementary Data 6).

### Cross-validation of lipid quantification
The TOF MS-based quantified values of selected lipid species in the NIST plasma SRM in both ionization modes were compared to those obtained via MRM measurements of the same extracts (Fig. 6 and Supplementary Data 6). Both instruments rendered similar lipid levels over the three quantification methods: 4D lipidomics in positive and in negative ion mode, and MRM, with average CV of <20% for the selected lipid species. This suggests that the selectivity of LC-TIMS-TOF-MS is comparably amenable for absolute quantification as MRM-selectivity. Cross-method concentration differences for individual lipids (CVs > 20%) are attributable to the superior structural resolution by 4D lipidomics compared to MRM afforded by ion mobility- and mass accuracy-based separation of co-eluting species that share identical fragmentation patterns (Fig. 6 and Supplementary Data 6).

The lipid levels from the NIST plasma SRM quantified with set 1 ISTDs and multi-point strategy were also compared to the consensus values reported in the NIST inter-laboratory study by Bowden et al.[21] (Supplementary Data 6). Most of the lipid species in our study have been identified in negative mode at the molecular species level, whereas in the study by Bowden et al. lipid species levels were determined[52]. Thus, quantitative differences between our 4D lipidomics platform- and inter-laboratory study are also attributed to a higher structural resolution afforded by 4D lipid descriptors. A similar comparative output was obtained for the values reported by (Wolrab et al., 2020). (Supplementary Data 6). The obtained lipid concentrations are in agreement with values reported by Bowden et al.[21] with CV values <30% for 70% of the lipids in both polarities, and also in agreement with values reported by Wolrab et al.[20] for HILIC-UHPLC-MS analysis, with CV values <30% for around 60% of the quantified lipid species in both polarities. The deeper coverage of plasma lipidome and its quantification is evident by the larger number of lipids quantified by the 4D lipidomics platform than currently reported in the inter-laboratory study and by Wolrab et al. (Supplementary Data 6).

### Applying 4D Lipidomics for intra-individual and multidien lipidome investigation in plasma, serum, blood, venous DBS, and finger-prick DBS (LBlooD study)
Intra-individual lipidome phenotyping in different blood matrices, amounting to a total of 180 samples, was reproducibly achieved using the 4D lipidomics quantification method with set 1 standards (Supplementary Data 9 and 10). As visualized in Fig. 7a by the radar plot of multidien lipidome for the five blood matrices of one individual, the

plasma and serum lipidome share a similar distribution pattern, and were found to be distinct from the rest of the blood matrices. The same holds true for DBS finger and DBS venous lipidome, which closely resemble each other, while, again, distinct from blood ethylene diamine tetraacetic acid (blood EDTA). The Wilcoxon signed-ranked test, performed on all features from both positive and negative mode analysis, confirmed the similarities and differences between lipidome of distinct blood matrices (Fig. 7b). Furthermore, the Principal Component Analysis (PCA) test using the molar concentration of lipids from both polarities (Fig. 7c) supports these findings wherein an overlap was observed between plasma and serum, and between DBS finger and DBS venous. The multiclass random forest classification, especially advantageous to evaluate differences between biological matrices with high number of features, resulted in an area under the receiver operating characteristic (AUROC) of 0.92, evidencing the presence of distinguishable features. The pairwise classification further supports the previous results (Supplementary Fig. 13). The AUROCs between DBS venous and DBS finger and between plasma and serum were in both ionisation modes marginally higher than random guessing. In contrast, the other AUROCs imply an almost perfect discriminability (Supplementary Fig. 14).

The next experiments were performed to learn how the individual lipidomes change over time. For reliable marker discovery and profiling, readily occurring diurnal changes of plasma lipidome, particularly of lipid mediators, are mitigated by standardized sampling procedures[53–55]. Intra-individual multidien change or stability of lipidome beyond the diurnal fluctuations is, in our opinion, an additional valuable attribute to factor in for selecting and qualifying lipid clinical markers. The overlap between the respective symbols of all the three time-points in the radar plot Fig. 7a indicated that the intra-individual lipidome was relatively constant over time. In addition, a Friedman test, chosen because the three time-points are dependent, was performed to understand the time-point variation in the lipidome. The output of the Friedman test evidenced no significant difference between the time points (Supplementary Fig. 15). The time-point variation of individual lipid classes in positive and negative ion mode (Supplementary Fig. 16) was also investigated for one of the individuals. The evaluation indicated a very low standard deviation (SD) between the time-points for most of the lipids. TG species exhibit highest multidien variation (≥20%) in plasma, serum, and in whole blood. This is in-line with the more dynamic bodily metabolic processing of TGs. Comparative assessment of levels of selected lipids determined by 4D lipidomics in negative and positive ion mode and by MRM (Supplementary Table 7) in all blood matrices of one individual (total 45 samples) showed CV values ≤25% for more than 70% of the lipids. This result emphasizes applicability of the method for reproducible lipidome phenotyping in various biological matrices and with different lipidome concentration pattern. Of note is that, despite acknowledged limitations of reversed-phase chromatography/MS for quantitative profiling[56,57], the quantification with the 4D lipidomics platform entailing reversed-phase chromatography permitted a clear-

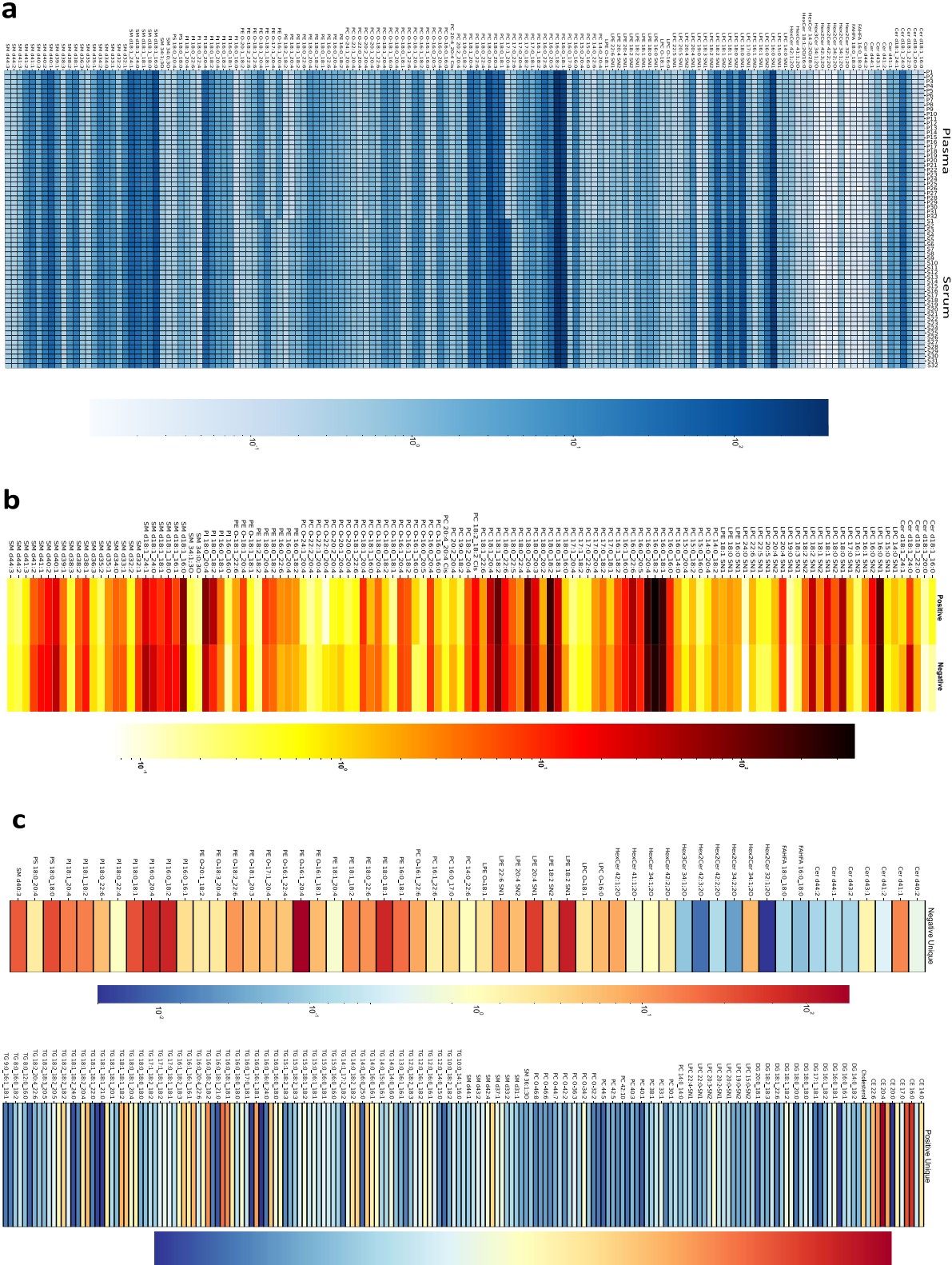

**Fig. 5 | Comparison and cross-validation of quantified lipid concentrations.**
**a** Heatmap visualization of the quantified values for 149 lipid species commonly identified between NIST human plasma and serum standard reference materials (SRMs) ($n = 32$ samples for each SRM). **b** Heatmap of the quantified values for 125 lipid species commonly identified in both ionization modes ($n = 32$ samples in each mode). **c** Ion mode unique lipid species. Heatmap visualization of the values obtained by the quantification of the lipids uniquely identified in negative (55) and positive (191) ion mode analysis of NIST plasma SRM ($n = 32$ samples in each mode). For each heatmap, $\log_{10}$ of the quantified values (nmol/mL) was used for the visualization. Source data are provided as a Source data file and in Supplementary Data 6.

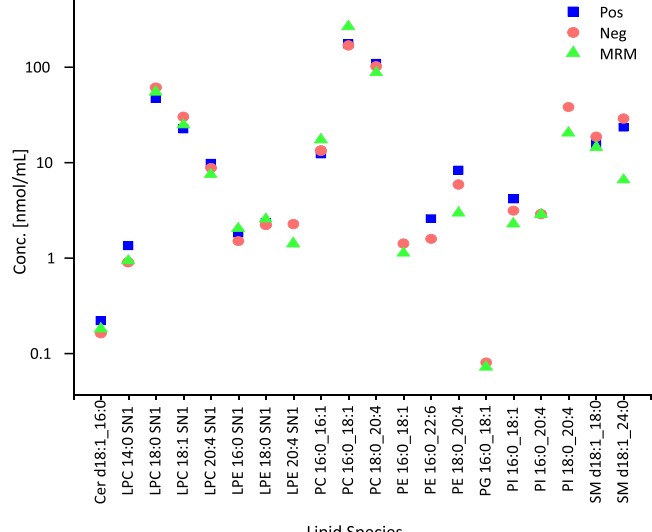

**Fig. 6 | Comparison and cross-validation of quantified lipid concentrations.** Comparison of the quantified lipid species between the positive and negative mode 4-Dimensional (4D) lipidomics ($n = 32$ samples in each mode) and multiple reaction monitoring (MRM) ($n = 10$ measurements) analysis. Source data are provided as a Source data file and in Supplementary Data 6.

cut delineation of the different blood matrix- and intra-individual lipidome phenotypes.

Noteworthy, CERT2 scores calculated from 4D lipidomics data of two individuals in LBlooD study showed consistent predictive values, of low risk, for all the three time points, additionally supporting the method's suitability for lipidome phenotyping and reliable clinical profiling (Supplementary Data 8).

## Discussion

The good sensitivity and reproducibility of the present high-throughput 4D lipidomics platform, along with confident lipid annotation and reproducible quantitative performance enable a deeper qualitative and quantitative plasma lipidome profiling than currently reported in inter-laboratory studies. Indeed, in addition to the quantification of 359 lipids out of 370 confident annotations, the selection of additional 470 unknown reproducible lipid features provides a valuable resource of unknown structures that can be prioritized for further structural elucidation. With the advancement in the procurement of lipid standards and generation of customized (sub)lipidome and subsequent 4D descriptors determination, the high-throughput 4D lipidomics platform can readily be expanded for a deeper lipidome coverage and quantification in preclinical and clinical research. It becomes also evident that as much as the availability of lipid standards affords, the choice of the ISTDs has to be tailored to the selectivity and sensitivity of the MS method as highlighted here for 4D lipidomics. Further systematic evaluation of the performance of all the available ISTDs and calibrants in certified lipid-striped matrices, as well as in reference samples of normo- and hyperlipidemia conditions will expedite selection of the best-suited standards for the 4D lipidomics platform. This is essential for the prospective harmonization of lipidomic protocols across an extended portfolio of MS methodologies to increase the quantification accuracy of endogenous lipids.

Certainly, the resolution of lipid isomers, particularly in high-throughput profiling with relatively short chromatographic gradients and broad ion mobility resolution, and subsequent quantification by 4D lipidomics platform has limits. Examples include: the resolution of sn1/sn2 phospholipids isomers is restricted to lyso-species, the position of the double bonds and differential sn- fatty

acyl substitution in glycerides, resolution of isobaric phospholipids with major differences in fatty acyl chains (PC 16:0/16:0 and PC 18:0_14:0, TG 16:0_18:1_20:4, and TG 18:1_18:2_18:2), etc. To overcome these limitations and extend lipidome coverage in complex biological matrices, development and integration of multi-analytical TIMS and PASEF-based protocols along with improved solutions for 4D data processing and visualization-based data mining and processing are required. Non-uniformity of lipidomic workflows across different MS strategies and laboratories is a recognized contributor to the variability of reported lipid levels for a given sample, potentially impeding consistent lipidome pheno-mapping in diseases, hence clinical translation. Despite this, the quantitative output of the high-throughput 4D lipidomics concurs well with that obtained by cross-platform and inter-laboratory analysis for the majority of commonly identified lipids. This consistent cross-platform lipidomic output is a solid basis to accelerate clinical translation of MS-based lipidomics, in general, including the high-throughput 4D lipidomics platform reported here. In this context, continuing the efforts initiated by Bowden et al.[20,21] to harmonize lipidomic methods, metrics, and outputs will be paramount. Given the lower resolution of ion mobility compared to reversed-phase chromatography, the TIMS-based separation alone did not aid in extending the lipid annotation coverage. CCS values are leveraged for increased annotation confidence and throughput, where CCS values serve as an added data attribute for confidence score, to confirm lipid isomer annotations and/or reduce misannotations and hits. With focus on the routine confident annotation in high-throughput profiling, where rather short chromatographic gradients are used, the added CCS data attribute was found useful for automatic and confident lipid annotation. Nevertheless, it is envisaged that future developments in mobilogram-based data processing tools will enable new lipid species annotations, while the provision of accurate CCS values for a broad set of lipid species and isomers will help infer annotations for IMS-based lipidomic platforms that use differing reversed-phase chromatographic gradients, chromatographic media or direct-infusion.

The high-throughput 4D lipidomics platform provides a reproducible qualitative and quantitative output for a large number of circulatory lipids, 359 in NIST plasma SRM and 354 in NIST serum SRM, as well as for intra-individual blood matrix lipidome pheno-mapping. Noteworthy, the consistent values obtained for the new predictive marker of cardiovascular disease and death and patient stratification, CERT2 score, for both NIST plasma SRM and intra-individual blood lipidome, highlight the viability of the method for high-throughput clinical profiling.

Newly revealed increased heritability of the plasma lipidome and the detailed description of the genetic association of plasma lipidome, evidence a stronger predictive power for disease risk of lipids other than traditional clinical lipid parameters[58]. Discovery of such candidate markers for disease prediction facilitated by lipidomics, and the successful discovery and implementation in clinical practice of CERT scores evidence the invaluable contribution of untargeted and targeted lipidomics for clinical marker discovery and profiling. Development of standard operating procedures for sampling, sample (pre) processing, storage, transport, extraction, and analysis, as well as operational mass spectrometric and data processing protocols[57,59], etc. are among the requisites to further advance clinical translation of lipidomic methods, including of 4D lipidomics platform presented here. This is essential to mitigate risks of bias or artifact marker discovery due to diurnal and multidien metabolic changes in lipidome, as well as pre-analytical and analytical parameters. In this regard, 4D lipidomics data gathered in this study highlight the contributing and limiting factors to achieve high confident lipid annotation, reproducible quantification in high-throughput mode, and can serve to inform and guide prospective study design for clinical translation of 4D lipidomics platform.

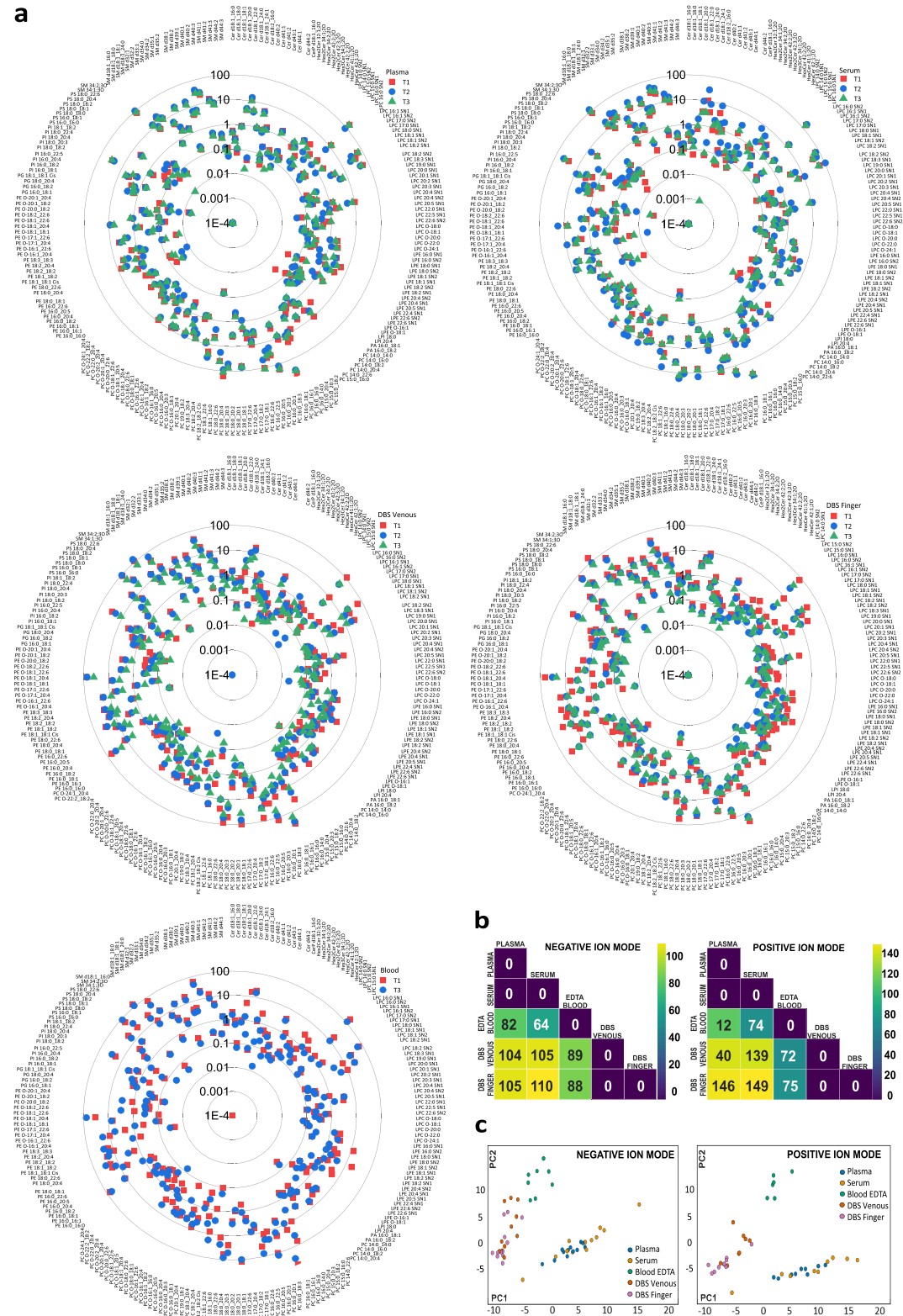

**Fig. 7 | Radar plot and statistical output for the quantified results from the negative mode in the LBlooD study. a** The radar plots represent the log$_{10}$ concentration of the quantified lipid species in all five biological matrices for an individual over the three-time points. **b** The two-sided Wilcoxon signed-rank test output shows the similarities and dissimilarities between different biological matrices for an individual in negative ion mode (left) and positive ion mode (right).

The color scale represents the index of similarity between any two biological matrices at a time with 0 representing similarity and the highest number representing the most dissimilarity. **c** Visualization of the first two principal components of the principal component analysis (PCA) to show the variability between biological matrices in negative ion mode (left) and positive ion mode (right). Source data are provided as a Source data file and in Supplementary Data 9 and 10.

Altogether, the good sensitivity, structural resolution, and reproducible quantification of plasma lipids afforded by the high-throughput 4D lipidomics platform support its suitability for future translation in the clinical discovery and profiling of lipid biomarkers in diseases.

## Methods

### Ethics statement

We herewith confirm that the LBlood study complied with all relevant ethical regulations regarding the use of human study participants and was conducted in accordance to the criteria set by the Declaration of Helsinki. For the LBlood study, four co-authors of the article, three females and one male with ages ranging from 26 to 60 years, consented to blood sample donation and analysis. Since no physicians are involved in our research project, the local ethics committee has no jurisdiction. Informed consent was obtained from each participant in the LBlood study and no compensation was provided. All participants in the study agreed with the provision of blood lipidome raw data in the data repository.

### Chemicals and solvents

The following LC-MS grade solvents and reagents used in analytical workflow were purchased from Merck (Germany): water, methanol (MeOH), 2-propanol, formic acid (FA), triethylamine, ammonium formate, HPLC-grade methyl tert-butyl ether (MTBE). Sodium hydroxide used in the preparation of recalibration solution was purchased from Thermo Fischer Scientific (Belgium). NIST plasma and serum SRMs were purchased from Merck (National Institute of Standards and Technology, USA).

### Lipid standards

**Calibration Standards.** 9-[(palmitoyl)hydroxy]-stearic acid (FAHFA 16:0/18:0), 1-hexadecanoyl-2-(9Z-octadecenoyl)-sn-glycero-3-phosphatidic acid (PA 16:0/18:1), 1-hexadecanoyl-2-(9Z-octadecenoyl)-sn-glycero-3-phosphocholine (PC 16:0/18:1), 1-hexadecanoyl-2-(9Z-octadecenoyl)-sn-glycero-3-phosphoethanolamine (PE 16:0/18:1), 1-stearoyl-2-linoleoyl-sn-glycero-3-phosphoethanolamine (PE 18:0/18:2), 1-hexadecanoyl-2-(9Z-octadecenoyl)-sn-glycero-3-phosphoglycerol (PG 16:0/18:1), 1-hexadecanoyl-2-(9Z-octadecenoyl)-sn-glycero-3-phosphoinositol (PI 16:0/18:1), 1-hexadecanoyl-2-(9Z-octadecenoyl)-sn-glycero-3-phosphoserine (PS 16:0/18:1), N-(hexadecanoyl)-sphing-4-enine-1-phosphocholine (SM d18:1/16:0), N-(hexadecanoyl)-sphing-4-enine (Cer d18:1/16:0), N-oleoyl-D-erythro-sphingosine (Cer d18:1/18:1), 1,2-di-(9Z-hexadecenoyl)-3-(9Z-octadecenoyl)-sn-glycerol (TG 16:1/16:1/18:1), 1,2-di-(9Z-octadecenoyl)-3-(5Z,8Z,11Z,14Z-eicosatetraenoyl)-sn-glycerol (TG 18:1/18:1/20:4), cholest-5-en-3β-ol (cholesterol), 1-hexadecanoyl-sn-glycero-3-phosphate (LPA 16:0), 1-octadecanoyl-sn-glycero-3-phosphoethanolamine (LPE 18:0), 1-octadecanoyl-sn-glycero-3-phosphocholine (LPC 18:0), 1-octadecanoyl-sn-glycero-3-phospho-(1′-myo-inositol) (LPI 18:0), 1-(9Z-octadecenoyl)-sn-glycero-3-phosphoserine (LPS 18:1), 1-hexadecanoyl-sn-glycero-3-phospho-(1′-sn-glycerol) (LPG 16:0) were obtained from Merck (Avanti Polar Lipids, Inc., USA).

**Internal standards (ISTDs for quantification).** 1-pentadecanoyl-2-oleoyl(d7)-sn-glycero-3-phosphate (PA-d7-15:0/18:1), 1-heptadecanoyl-2-(9Z-tetradecenoyl)-sn-glycero-3-phosphocholine (PC 17:0/14:1), 1-heptadecanoyl-2-(9Z-tetradecenoyl)-sn-glycero-3-phosphoethanolamine (PE 17:0/14:1), 1-heptadecanoyl-2-(9Z-tetradecenoyl)-sn-glycero-3-phosphoglycerol (PG 17:0/14:1), 1-heptadecanoyl-2-(9Z-tetradecenoyl)-sn-glycero-3-phosphoinositol (PI 17:0/14:1), 1-heptadecanoyl-2-(9Z-tetradecenoyl)-sn-glycero-3-phosphoserine (PS 17:0/14:1), N-(heptadecanoyl)-sphing-4-enine (Cer d18:1/17:0), N-(dodecanoyl)-sphing-4-enine-1-phosphocholine (SM d18:1/17:0), cholest-5-en-3β-ol(d7) (cholesterol d7), 1-heptadecanoyl-glycero-3-phosphate (LPA 17:0), 1-(10Z-heptadecenoyl)-sn-glycero-3-phosphocholine (LPC 17:1), 1-(9Z-heptadecenoyl)-sn-glycero-3-phosphoethanolamine (LPE 17:1), 1-(9Z-heptadecenoyl)-glycero-3-phospho-(1′-sn-glycerol) (LPG 17:1), 1-(10Z-heptadecenoyl)-sn-glycero-3-phospho-(1′-myo-inositol) (LPI 17:1), 1-(9Z-heptadecenoyl)-glycero-3-phosphoserine (LPS 17:1), N-palmitoyl-D-erythro-sphingosine (d7) (Cer d18:1-d7/16:0), N-stearoyl-D-erythro-sphingosine (d7) (Cer d18:1-d7/18:0), N-lignoceroyl-D-erythro-sphingosine (d7) (Cer d18:1-d7/24:0), (N-hexadecenoyl-D-erythro-sphingosylphosphorylcholine-d9 ((SM d18:1/16:1)-d9), (N-oleoyl-D-erythro-sphingosylphosphorylcholine-d9 ((SM d18:1/18:1)-d9), (N-Nervonoyl-D-erythro-sphingosylphosphorylcholine-d9 ((SM d18:1/24:1)-d9), 5-[((13,13,14,14,15,15,16,16,16-d9)palmitoyl)hydroxy]-stearic acid (FAHFA 16:0/18:0 d9), 9-[((13,13,14,14,15,15,16,16,16-d9)palmitoyl)hydroxy]-stearic acid (FAHFA 16:0/18:0 d9), 1-heptadecanoyl-2-hydroxy-sn-glycero(d5)-3-phosphocholine (LPC 17:0 d5), 1-heptadecanoyl-2-myristoleoyl-sn-glycero(d5)-3-phosphocholine (PC 17:0/14:1 d5), 1-heptadecanoyl-2-oleoyl-sn-glycero(d5)-3-phosphocholine (PC 17:0/18:1 d5), 1-heptadecanoyl-2-eicosatrienoyl-sn-glycero(d5)-3-phosphocholine (PC 17:0/20:3 d5), 1-heptadecanoyl-2-hydroxy-sn-glycero(d5)-3-phosphoethanolamine (LPE 17:0 d5), 1-heptadecanoyl-2-myristoleoyl-sn-glycero(d5)-3-phosphoethanolamine (PE 17:0/14:1 d5), 1-heptadecanoyl-2-oleoyl-sn-glycero(d5)-3-phosphoethanolamine (PE 17:0/18:1 d5), 1-heptadecanoyl-2-myristoleoyl-sn-glycero(d5)-3-phosphoinositol (ammonium salt) (PI 17:0/14:1 d5), 1-heptadecanoyl-2-oleoyl-sn-glycero(d5)-3-phosphoinositol (ammonium salt) (PI 17:0/18:1 d5), 1-heptadecanoyl-2-docosatetraenoyl-sn-glycero(d5)-3-phosphoinositol (ammonium salt) (PI 17:0/22:4 d5), 1-heptadecanoyl-2-myristoleoyl-sn-glycero(d5)-3-phospho-L-serine (sodium salt) (PS 17:0/14:1 d5), 1-heptadecanoyl-2-oleoyl-sn-glycero(d5)-3-phospho-L-serine (sodium salt) (PS 17:0/18:1 d5), 1-heptadecanoyl-2-myristoleoyl-sn-glycero(d5)-3-phospho-(1′-rac-glycerol) (sodium salt) (PG 17:0/14:1 d5), 1-heptadecanoyl-2-oleoyl-sn-glycero(d5)-3-phospho-(1′-rac-glycerol) (sodium salt) (PG 17:0/18:1 d5), 1,3(d5)-ditetradecanoyl-2-(9Z-hexadecenoyl)-glycerol (TG 14:0/16:1/14:0 d5), 1,3-diheptadecanoyl-2-(10Z-heptadecenoyl)-glycerol (d5) (TG 17:0/17:1/17:0 d5), and 1,3(d5)-dinonadecanoyl-2-dodecanoyl-glycerol (TG 19:0/12:0/19:0 d5) were also obtained from Merck (Avanti Polar Lipids, Inc., USA).

**Standards for the in-house library.** 200 lipid standards covering all the major phospholipid classes such as glycerophospholipids, TGs, DGs, sphingolipids, and cholesterol for the generation of the in-house library were purchased from Merck (Avanti Polar Lipids, Inc., USA).

### Automatic high-throughput extraction platform

To optimize the extraction efficiency of lipids, a liquid–liquid extraction (LLE) method as described in ref. [22] based on MTBE/methanol (10:3; v/v)[23] was adapted to high-throughput extraction, using the epMotion® 5070 liquid handling workstation. NIST plasma SRM, as well as NIST serum SRM, were used for the optimization of lipid extraction and for the generation of replicate lipid extracts for 4D PASEF method development. Therefore, 20 µl aliquots of each (plasma and serum) were pipetted into a 2 ml deepwell plate (Eppendorf, Germany) and the plate was placed into the epMotion plate holder where all further extractions steps were carried out by the epMotion workstation (Supplementary Data 11 and 12). 615 µL cold MTBE, 185 µl cold MeOH, containing the spiking solution, as well as 250 µL cold FA (0.1%) was added to the samples. After 10 min vortexing of the samples in the workstation (8 °C and 1050 revolutions per minute (rpm)), the 2 ml deepwell plate was centrifuged for a further 10 min (5000 × g swing out/4 °C). After placing the 2 ml deepwell plate back in the epMotion, the upper organic phase was transferred to a fresh 1 ml deepwell plate (Eppendorf, Germany) and subsequently evaporated under a gentle stream of nitrogen at 37 °C in the Vapotherm (Barkey, Germany). Lipid extracts were reconstituted in 360 µL methanol/water (9:1; v/v) and transferred to a twinTec PCR injectplate (Eppendorf, Germany) for further LC-MS analysis. The remaining lipid extracts were

evaporated to dryness and stored at −20 °C till further analysis. To minimize ex vivo alterations of the endogenous lipid levels, all extraction procedure steps, where implementable, were carried out at 4 °C. For evaluation of the lipid extraction efficiency via the automatic platform, the 64 NIST plasma SRM samples from the inter-day extraction experiment spiked with set 2 ISTDs (see below and Supplementary Table 4) were compared to 8 samples of the same NIST plasma SRM aliquot spiked with the same set of ISTDs after extraction. For matrix effect assessment, the 8 NIST plasma SRM samples spiked after extraction were compared to set 2 ISTDs without plasma matrix ($n = 21$). To evaluate the performance of the automized high-throughput extraction method in comparison to a manual workflow, lipids from the same NIST plasma SRM aliquot ($n = 8$) were manually extracted with the set 2 ISTDs, using identical extraction solvents. The manual extraction was conducted in brown 1.5 ml Eppendorf tubes. The samples were vortexed for 10 min in a thermocycler (4 °C and 1200 rpm) and centrifuged for 10 min at 13,000 rpm and 4 °C[22,48]. The recovery and matrix effects of the manual workflow were assessed using 8, additional, NIST plasma SRM aliquots that were manually extracted and spiked after the extraction with set 2 ISTDs and compared to set 2 ISTDs without plasma matrix ($n = 8$). For the No-extraction, 20 μl of NIST plasma SRM ($n = 8$) were diluted 1:20 and spiked with set 2 ISTDs to the same final concentration as for the automated and manual extraction, respectively.

### Liquid chromatography

For negative ion mode, LC separation was achieved with Elute UHPLC (Bruker Daltonics, Germany) using mobile phase A: methanol/water (1:1; v/v) and mobile phase B: methanol/isopropanol (2:8; v/v), both containing 0.1% formic acid, 7.5 mM ammonium formate, and 0.1% triethylamine (TEA), respectively. For positive ion mode, the same solvent system was used without TEA. The LC method consisted of a 20 min run time with the following gradient: T0 to T1 – mobile phase A at 60% and mobile phase B at 40% which was then elevated to 90% mobile phase B and 10% mobile phase by T16 and up to 99% mobile phase B by T16.5 and then escalated back to 40% mobile phase B by T20. A constant flow rate of 0.2 mL/min was maintained throughout the runtime. Throughout the analysis, the autosampler was maintained at 4 °C. The samples were injected onto the column with a partial loop mode with an injection volume of 20 μL and 10 μL corresponding on the column to 1 μL plasma in negative ion mode and to 0.5 μl plasma in positive ion mode, respectively. For the separation, a C18 Luna Omega column (100 × 2.1 mm × 1.6 μm) (Phenomenex, Germany) was used and thermostated at 45 °C. Total run time including column wash was 20 min. No carry-over was observed in negative and positive ion mode when using wash-run protocol and analysis batch structure inferred from previous work[22,59].

### MS acquisition

**4D PASEF.** The 4D PASEF experiments were acquired using a TIMS-TOF pro instrument for the negative ion mode and TIMS-TOF flex operating in electrospray ionization mode for the positive ion mode (Bruker Daltonics, Germany). The parameters were optimized using standard lipid mixtures and plasma extracts and set as follows: (a) Source parameters: End plate offset was set to 500 V, while the capillary voltage at 4500 V for positive ion mode and to 3600 V for negative ion mode. The nitrogen as the dry gas was run with the rate of 6 L/min at dry temperature of 200 °C. The nebulizer gas was set at 2.5 bar. The peak detection threshold was set to 100 counts. Scan mode was set to PASEF with the mass scan range of 100 to 1350 Da for both MS and MS[2] acquisition. The acquisition cycle consisted of 0.1 s with the mobility scan range of 0.55 to 1.87 V*s/cm[2] for the positive mode and 0.55 to 1.86 V*s/cm[2] for the negative mode. The collision energy was set to 35 eV in positive and 45 eV in negative ion mode. Both the TIMS and mass calibration of the instrument was carried out on a

weekly basis with the following peaks from the Agilent ESI LC-MS tuning mix [$m/z$, $1/K0$: (322.0481, 0.7318 Vs cm$^{-2}$), (622.0289, 0.9848 Vs cm$^{-2}$), (922.0097, 1.1895 Vs cm$^{-2}$), (1221.9906, 1.3820 Vs cm$^{-2}$)] in the positive mode, and [$m/z$, $1/K0$: (666.01879, 1.0371 Vs cm$^{-2}$), (965.9996, 1.2255 Vs cm$^{-2}$), (1265.9809, 1.3785 Vs cm$^{-2}$)] in the negative mode. An online recalibration of the data was performed immediately after each sample acquisition using a mixture of Agilent ESI LC-MS tune mix and 1 mM sodium formate (1:1) injected directly into the ESI source via a syringe pump. For this, the 20 min LC-MS runtime was divided into three segments with the first segment (0.0 to 0.05 min) used for method equilibration, the second segment (0.05 min to 0.3 min) used for the injection of tune mix-sodium formate, and the last segment (0.3 min to 20 min) used for the sample data acquisition. The switch between the segments was achieved using a conventional 6-port divert valve. The peaks list from this mixture used for the recalibration is presented in Supplementary Data 13. Extensive formation of sodium adducts of lipids, particularly in positive ion mode was detected when factory-recommended concentration of sodium formate calibrant was used for online recalibration. Therefore, the concentration of sodium formate was reduced from 10 mM to 1 mM without compromising the performance.

**MRM.** For the inter-instrument cross-validation of the plasma quantification, targeted LC-MRM analyses were performed using a 5500 QTrap triple-quadrupole linear ion trap mass spectrometer equipped with a Turbo V Ion Source (AB SCIEX, Darmstadt, Germany) with polarity switching. The LC, ionization, detection, and MRM transition parameters for selected lipids were set as previously described[22,48] and are provided as a Source data file.

### Generation of the in-house library and analyte list

The 200 lipid standards purchased from Avanti Polar Lipids were screened individually in triplicates and in both ionization modes to retrieve for each of the ion types: the CCS and RT values, and fragmentation spectra. The in-house library in both polarities was generated by adding the MS[2] spectra of one of these triplicates. The CCS and RT values for the individual lipid standards were calculated by taking an average of the values from the triplicates and adding them to the library. The complete list of all these standards can be found in Supplementary Data 2. The in-house library was further expanded by the addition of lipid species annotated and curated from the bucket list consisting of multiple analyses of NIST plasma SRM lipid extracts (Supplementary Data 3). For this, the NIST plasma SRM lipids were in the first place identified with the aid of the MSDIAL-TandemMassSpectralAtlas-VS68 positive and negative spectral database. To investigate the origin of hereby identified lipids and select the reliable ones, we performed a NIST plasma SRM dilution experiment in triplicates, starting with 2 μl plasma on the column and sequentially diluted in five 1:2 steps, respectively, up to 0.0625 μl plasma on the column. Features not exhibiting a plasma dilution response, as determined in Metaboscape 2021a (Bruker Daltonics, Germany) Box Plot were manually removed. The remaining identified lipid species from the bucket list were additionally manually curated using the rule-based fragmentation of the lipids, their corresponding characteristic MS[2] fragment spectra, and the comparison of the RT and CCS values generated by the analysis of the 200 purchased standards (Fig. 3 and Supplementary Data 2 and Supplementary Figs. 17–20). Spectral matching to MS DIAL and manual curation were used for the initial curation of lipid species annotation in plasma and establishment of plasma reference database as exemplified for PC-O 24:2_20:4 (Supplementary Fig. 17). Diagnostic fragment ions for discrimination of FAHFA from possible fatty acyl dimers were used to curate these species from plasma[60], in addition to RT and CCS reference of the standard (Supplementary Fig. 18). In cases where additional fragments, corresponding to an isobaric structure, in an MS[2] spectrum were

identified on manual curation, the isobar with the most abundant alkyl fragments in that MS$^2$ spectrum was selected for the annotation of that particular feature (Supplementary Figs. 19 and 20). Furthermore, the analyte lists (ClinLip Analyte lists) for each polarity were created to also have the RT, CCS, and neutral mass information for all the lipid species present in the expanded library. This was achieved by linking the expanded spectral library in the respective ClinLip analyte lists (one for each polarity) to have all four dimensions, RT, CCS, *m/z*, and fragmentation spectra in one annotation source. All the annotations were thereby performed only with the ClinLip Analyte list of the matching polarity (Supplementary Data 3).

**Identification.** To expedite routine and confident lipid identification and subsequent quantification, we performed a repetition experiment, comprising the extraction of lipids from 32 NIST plasma and serum SRM samples (a 20 μl aliquot each) using the established automatic extraction method and subsequent 4D TIMS profiling. To evaluate inter-assay reproducibility, all the plasma extracts were remeasured after two weeks amounting to a total of 64 replicates. For lipid assignment, the obtained data were transferred to Metaboscape. Feature detection was performed using an intensity threshold of 200 counts and the rules to filter the 4D extracted features by the number of occurrences were defined as (1): recursive feature extraction is performed for features found in at least 17 out of 32 analyses, and (2): the feature is included in the bucket table only if found in 32 out of 32 analyses after recursive feature extraction. Following that step, a background subtraction of the features was conducted, using a MeOH/ isopropanol sample analysis. The in-house acquired ClinLip Analyte list was used for subsequent lipid annotation. The scoring system for high-confidence lipid species identification is based on the 4D descriptors: mass accuracy, RT, CCS, and MS$^2$ spectra. The parameter ranges were determined by evaluating the reproducibility of the descriptors in the intra- and inter-day assay experiments and were set as follows: mass accuracy 1–3 ppm/RT 0.1–0.5 min/ CCS 0.2–1.5 A$^{o2}$/MS/MS score 900-500 (the score is compounded by fit, reverse fit, and purity comparison to the spectral library, with 900 being the limit for good confident scores, and 500 is the lower cut-off for moderate score).

The identification of the stable unknown features from the repetition experiment was performed in two major steps. The first step involved finding the overlap of the unknown features between the repetition and the dilution experiment. The overlap between features measured in the two experiments was defined in terms of similarity in the *m/z*, retention time, and CCS values. We, therefore, defined the tolerance values as: <0.002 *m/z*, 0.1 min for RT, and 0.2 A$^{o2}$ CCS. This part was accomplished via a Python script using NumPy (1.22.0). The identified overlapping features were then also controlled for their dilution response using the data obtained in the dilution experiment. The features were accepted when satisfying the following two criteria: (1) the value of the feature was positively correlated with the dilution using the Pearson correlation ($p \geq 0.9$) and (2) the SD from the mean of the dilution experiments was >0.1. Following this automated analysis, the list of identified overlapping features was manually curated and validated (Fig. 3). To evaluate if stable unknown features correspond to lipid species comprised in the metabolome database, we also subjected the replicate plasma extracts to Metabobase annotation, which however, did not provide any annotation hint.

## Quantification strategies

For the method development, assessment, and phenotype identification of LBlooD samples the set 1 of standards comprising class-specific exogenous lipids not commonly found in the human plasma as ISTDs and deuterated ISTDs for lipid classes such as PA, cholesterol, and TGs was used (Supplementary Table 3). For inter-plate assay (64 plasma samples extracted over 2 plate) and assessment of ISTDs influence on

quantification output, a set 2 of standards comprising deuterated ISTDs was used (Supplementary Table 4).

Two different strategies namely: multi-point and one-point quantification were assessed. The quantification was performed by retrieving the peak areas of the corresponding lipids from Metaboscape to an excel file wherein the quantification was performed using a base sheet. Peak area values determined by the recursive algorithm of Metaboscape were invariably outliers (were found to be incongruent with Compass Quantanalysis) and thus, not considered for quantification. Invariably, the acceptance criterion for the accuracy of calculated concentration was set to ±20%.

**Multi-point.** For the multi-point strategy, external standards (https://lipidomicstandards.org/lipid-species-quantification/)[47] were serially diluted to 7 concentration points (calibration points) and each of the 7 calibration points spiked with specific ISTDs. For this, first, the linear concentration range for each lipid class was determined (Fig. 4a) using set 1 standards (Supplementary Table 3). Based on the response of individual lipid classes over the tested concentration range, class-specific calibration curves were set for the multi-point quantification strategy. A calibration curve was constructed by plotting the ratio of peak areas of the external standards to the peak areas of ISTDs as a function of the ratio of the concentration of external standard to the concentration of ISTDs [$(A_{ES}/A_{IS})/(C_{ES}/C_{IS})$]. The regression coefficients 1 and 2 were then used to calculate the concentration of the lipids in the samples (see Supplementary Data 14 for regression parameters).

**One-point.** For the one-point strategy, a single concentration point of a class-specific external standard (one-point a) spiked with an ISTD was used. For one-point quantification, a parameter (m) was obtained by dividing the ratio of peak area of external standard normalized to the peak area of ISTD to the concentration of ES normalized to the concentration of ISTD [$(A_{ES}/A_{IS})/(C_{ES}/C_{IS})$]. This parameter was used for the quantification of lipid species using one-point quantification.

Multi-point quantification:

$$A = \left( \left( \frac{\frac{\text{Peak area of analyte}}{\text{Peak area of ISTD}}}{\text{regression coefficient} - 1} \right)^{\text{regression coefficient} - 2} \right) * (\text{Conc. of ISTD}) \quad (1)$$

where $A$ = concentration on column (ng mL$^{-1}$).

Regression coefficient-1&2 obtained from Cal. curve:

$$B = \left( (A) * \frac{\text{Final Volume after re} - \text{dissolution (μL)}}{\text{Volume of sample used for extraction (μL)}} \right) \quad (2)$$

where $B$ = normalized concentration in sample (ng mL$^{-1}$).

$$C = \frac{B}{\text{Measured adduct mass (Da)}} \quad (3)$$

where $C$ = concentration in whole sample (nmol mL$^{-1}$).

One-point quantification:

$$m = \frac{\frac{\text{Peak area of ESTD}}{\text{Peak area of ISTD}}}{\frac{\text{Conc. of ESTD}}{\text{Conc. of ISTD}}} \quad (4)$$

where $m$ = Calibration parameter.

$$A = \left( \left( \frac{\frac{\text{Peak area of analyte}}{\text{Peak area of ISTD}}}{m} \right) \right) * (\text{Conc. of ISTD}) \quad (5)$$

where $A$ = Concentration on column (ng mL$^{-1}$). $B$ & $C$ remains the same as that for multi-point quantification.

The empirical relation used for the multi-point and the one-point quantification, respectively, is shown above. The acquisition sequence for any of the experiments contained at least three sets of 7-point calibration standards typically at the beginning, middle, and end of the batch/sequence, and a minimum of three QC (Supplementary Table 8) samples within the sequence, with one QC being invariably acquired at the beginning of the sequence for system suitability test. The acceptance criterion for each point in the calibration curve was set to ±20% for accuracy.

**Quantification and detection performance.** Limit of detection (LLOD) and limit of quantification (LLOQ) for each lipid class in both positive and negative polarities were determined following the guidelines for bioanalytical method validation[17,39,61]. For each lipid class, a representative class-specific external standard was serially diluted over 7 linear points of the tested concentration range and each of the 7 points was spiked with a level-3 class-specific ISTD. The parameters used for the calculation of the LLOD and LLOQ, with the empirical relation shown in the adjacent box, were determined by the analysis of variance (ANOVA) statistical test. The intra-day and inter-day re-measurement validation of the multi-point quantification was performed by applying it to a data set ($n$ = 32 extracts) acquired in one batch and remeasuring the $n$ = 32 extracts after two weeks for inter-day assay. Validation of inter-plate assay using multi-point quantification strategy was performed using 64 plasma samples extracted over two plates ($n$ = 32 samples per plate, 2 plates) with the use of set 2 standards (Supplementary Table 4). Linearity parameters are listed in (Supplementary Data 14).

Method Validation:

$$LLOD = 3.3\frac{\sigma}{s} \qquad (6)$$

$$LLOQ = 10\frac{\sigma}{s} \qquad (7)$$

Where LLOD = lower limit of detection; LLOQ = lower limit of quantification; $\sigma$ = standard deviation of responses; s = slope of calibration curve.

## LBlooD study

For the demonstration of the applicability of the analytical method and quantification strategy, we decided to identify and quantify lipid species from different blood matrices collected over three-time points from four author volunteers. This pilot sample cohort was chosen as representative for various lipidome-phenotypes due to expected differences between the different blood matrices. Hence, it served as a tool to evaluate the amenability of the 4D TIMS-based annotation and subsequent quantification methods to reproducibly identify and quantify lipids in the different blood matrices and reproducibly identify differences between individual blood matrix lipidome phenotypes within an individual.

The three-time points selected for this purpose were: 0-day, 1-week, and 1-month after the first collection. The different biological matrices selected for this purpose were: plasma, serum, blood, DBS from the vein, and DBS from the fingertip blood. For this, two blood samples, one in 9 mL EDTA and the other in 7.5 mL serum monovette, were collected from four authors of this study. For the plasma and serum analysis, the serum monovette was centrifuged at 4 °C, 2000 × $g$ for 10 min directly after the blood was collected. The resulting upper plasma/serum phase pooled and transferred to 5 mL brown Eppendorf tubes. From this pooled volume, three 20 μL aliquots were collected into three 1.5 mL brown Eppendorf tubes (to avoid light-induced

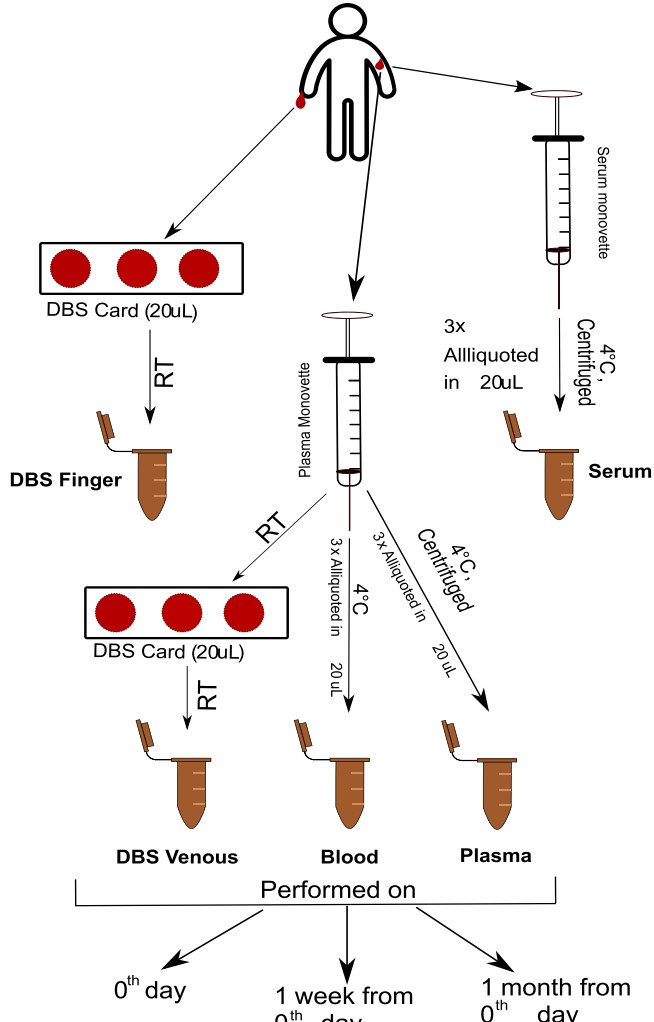

**Fig. 8 | Workflow of sample collection for the LBlooD study.** Schematic representation of the sampling and pre-processing strategy for lipid extraction from five different biological matrices in an individual. The same extraction procedure was followed for sample preparation on the 0th day, 1 week from the 0th day, and 1 month from the 0th day. This procedure was followed for the sample collection from all participating individuals in the study.

chemical changes). The tubes were then stored at −80 °C until the extraction. For the whole blood analysis, before the centrifugation process, 20 μL of blood from the EDTA tube was collected into 3 brown Eppendorf tubes each and placed on ice until the extraction (which was performed on the same day). For the DBS venous, 20 μL of blood from the EDTA tube was spotted in the center of each three spots of a DBS card (GE Healthcare Bio-sciences, USA). A similar procedure was followed for the DBS finger except that the blood used for the spot was from the fingertip. The DBSs were dried for 3–4 h at room temperature until the spots have a homogeneous brown color and then stored at −20 °C until the further procedure. Just before the extraction, the whole DBS was cut along the broken lines on the edges of the spot and each spot tore into 4 pieces and directly added to the brown Eppendorf tubes. The entire sample preparation workflow for this experiment has been illustrated in Fig. 8. All the blood matrices were extracted using the MTBE/MeOH extraction protocol[48].

## Quantification of NIST SRMs and the LBlooD study

All the NIST plasma and serum SRM lipid extracts, respectively, as well as samples from LBlooD study were quantified using the multi-point quantification strategy and level-3 ISTDs. 32 NIST plasma SRM and

serum samples, respectively, were quantified with set 1 standards. Additional $n = 64$ NIST plasma SRM samples, extracted over two plates, were quantified using set 2 standards to asses inter-plate assay reproducibility and compare performance of ISTDs/calibrants. The same procedure was followed for the quantification of the LBlooD study using set 1 standards with the exception that the class-specific ISTD (SM d18:1_12:0) was used for the manually extracted SM lipid class in both ion modes. For LBlooD study, 45 samples per each individual were obtained and analysed in both ion modes, amounting to a total of 180 samples from study participants per ionization mode. The quantification of the 4D lipid data in both polarities from the TIMS-TOF survey data was performed using the multi-point strategy. For multi-dien lipidome stability assessment, the quantified values of all the identified lipids were compared between the matrices in each individual over all the three-time points. For additional validation, MRM measurements and quantification of selected lipids for one individual were performed as described above and in refs. [22,48].

## Software

The instrument calibrations and the data acquisition were controlled by Compass Hystar 6.0 and timsControl 2. Data processing including identification and annotation of the lipids using the filters mentioned above was performed using Metaboscape 2021a. After the identification via the ClinLip Analyte list, the annotated bucket table was exported to the MS excel 2019 and all the quantifications were performed therein. The data processing for bioanalytical method validation was performed using Compass Quantanalysis v5.3. All the visualizations and statistics were produced using MS Office Professional Plus 2019, Python 3.9.7 (Spyder 5.1.5, PyCharm 2021.1, Jupyter-notebook 6.2.0, NumPy 1.20.2, SciPy 1.6.3, Scikit-learn 0.24.2), Inkscape (V5), and Origin 2021.

## Statistical analysis

For the assessment of the intra-individual lipid differences and similarities across time-points (multidien) and different blood matrices following statistical analyses were performed and a $p$-value of <0.05 was considered significant:

(1) A Wilcoxon signed-ranked test has been performed to analyse the differences in the lipid intensity between blood matrices. In this experiment, the data for every individual at all time points of one blood matrix were used as one data set. All blood types were compared to each other. The comparisons were computed individually for every lipid and the number of times $H_0$ got rejected was counted.

(2) A Friedman test was performed for the assessment of multidien lipidome changes: For this test, the individual information was omitted and the samples of the individuals at one-time point were considered as one data set. Again, it was counted how often $H_0$ got rejected. The Benjamini–Hochberg method was applied on the $p$-values of the Friedman and Wilcoxon tests to account for spurious significant results due to multiple testing.

(3) Random forest classification: The/A multiclass and pairwise classification was performed for a thorough analysis and comparison, especially with respect to test metrics and runtime, for both multiclass and one vs rest classification methods. The pairwise classification determines explicitly the distinguishability between two matrices similar to PCA analysis. While PCA provides an intuitive visualization, it cannot be used directly for the mentioned tasks. PCA is a dimension-reduction tool, but not a classifier. The random forests were trained with a $3 \times 4$ cross-validation, where each fold contained the data samples of only one person. This avoids information leakage from the training data to test data, which would occur if data samples of one individual would be in the training and the test set at the same time. For multiclass classification, one random forest was trained

to classify samples into one of the five possible biological matrices, and the area under the receiver operating characteristic (AUROC) was computed as test metric. The multiclass classification gives an overall expression of how distinguishable the matrices are, but not for the individual comparison between two types. In addition, a pairwise classification, where four different random forests were trained for every matrix to distinguish its samples individually from every other matrix, was performed. The features of one matrix were used as positive class and the features of another matrix as negative class. This was repeated for every possible pair. The insights can also be used on a larger number of features (180 samples × 364 lipids) to distinguish varying conditions and/or the corresponding blood matrix. However, the current data set was not used as a subject data for blood matrix prediction with random forest classification.

(4) Principal component analysis (PCA) was performed to evaluate the similarities between biological matrices. The first two components were employed to create a two-dimensional visualization of the samples.

(5) Pooled Variance: The variances of the lipids over all the three-time points for each biological matrix were calculated. The pooled variance of all lipids of one lipid class was computed and the square root was used to derive the SD. The pooled SD represents mean-variance over the time points in one lipid class.

## Reporting summary

Further information on research design is available in the Nature Portfolio Reporting Summary linked to this article.

## Data availability

The mass spectrometric raw data generated in this study have been deposited in the Metabolomics workbench data repository (supported by NIH grant U2C-DK119886)[62] with study ID ST002402 [https://doi.org/10.21228/M88T45]. Source data are provided with this paper.

## Code availability

Python scripts used to generate the statistics used in the publication is available at https://github.com/kramerlab/lipid_study/.

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

## Acknowledgements

This work was financially supported by the BMBF-funded DIASyM Nr: 031 L0217 A project TP3 to L.B. R.L., D.B., and C.S. are financed by the BMBF-funded Project Nr: 031 L0217 A, project TP3 to L.B. J.M.P. is financially supported by the SPP 2225 project nr Bl 1399/2-1 to L.B. We thank Dr. Sven W. Meyer for the assistance with data processing. D.B. is registered PhD student at TransMed program. We thank Univ. Prof. Dr. Philip Wild from Center for Cardiology at University Medical Center Mainz for the invaluable insights into the requirements for clinical translation of mass spectrometry.

## Author contributions
R.L. and D.B. performed the experiments and data analysis and optimized protocols. C.S. contributed technical support to implementation of automatized extraction and routine instrument maintenance. J.M.P. contributed to data processing and visualization. S.N. and T.H. performed the computational studies and statistical analysis. S.K. supervised the statistical analysis. R.L., D.B., and L.B. interpreted the results and wrote the manuscript. L.B. conceived and coordinated the study and acquired funding.

## Funding

## Competing interests
The authors declare no competing interests.
