## [Peer Review File · Nature Communications]

REVIEWER COMMENTS

Reviewer #1 (Remarks to the Author):

The manuscript by Lerner et al., provides a nice insight into the utility of PASEF technology for clinical lipidomics. The manuscript nicely outlines various steps undertaken to validate and benchmark the technology for such analyses. Furthermore, the quality of the lipidomics data appears to be better than that of Vasilopoulou et al. (2020); which is very good and seems to underscore that also PASEF technology can be used for screening purposes in clinical lipidomics.

Nevertheless, the manuscript and the work have a range of issues that can be addressed. First, the overall readability of the manuscript can be improved. As it stands, there is a lot of effort devoted to using buzzwords rather than articulating more concisely and exactly how the results were obtained. Notably, the authors somewhat inflate the added value of the TIMS dimension in lipid annotation and quantification. Moreover, from reading the main text it is not clear whether PASEF data (i.e., TIMS-TOF MS/MS data) is used for quantification or whether survey TOF MS data is used for quantification. Here, the latter seems to be the case albeit the authors use the then inappropriate term "4D-PASEF quantification".

Second, it would be outstanding if the authors had devoted more effort to assess the analytical sensitivity and dynamic quantification range of the platform using pairs of lipid class-specific, stable isotope-labelled standards spiked into plasma samples, followed by lipid extraction and MS analysis. This should be done for the majority of lipid classes already assessed by the multi-point strategy (Fig. 2a) as well as cholesterol, CE, TG and DG lipids. Plus, it should be done using plasma matrices with hyperlipidemia and normolipidemia. Dynamic quantification range curves (based on non-smoothed not regression-corrected data!) can be presented as supplementary information; and LOD and LOQ values can be added next to Fig. 2a and 2b. Moreover, LOD and LOQ values should be compared to that obtained by the LC-MRM analysis. The point here is, that the authors insinuate over and over again that the 4D-lipidomics platform is very sensitive, but provide no data to support this claim. I believe that the sensitivity of the 4D-lipidomics platform is comparable to that of most other lipidomic platforms, including those based on standard QqTOF and Orbitrap technology. Thus, the authors might eventually want to downplay the hype about the high sensitivity of the timsTOF technology.

Third, the overall efficacy of the 4D-lipidomics platform is highlighted by the ability of the platform to accurately discriminate whether analyzed blood samples were obtained as plasma, serum, whole blood, venous DBS and finger-prick DBS. Here, it would have been more interesting, and relevant,

had the utility of the platform been showcased by analyzing a limited cohort of, for example, ~50 subjects with hyperlipidemia and normolipidemia.

Finally, I have attached an annotated pdf file for the authors to try to use to improve their manuscript. This pdf file features general comments and suggestions, emphasized by a red vertical line on the right.

Other important comments (not exhaustive):

-It is awkward, and almost mysterious, why the authors do not detect any FA chain isomers (e.g., Supplementary Data 8 Analyte List). For example, it cannot be that the authors only detect PC 18:0-18:2 in human plasma. There must be some PC 18:1-18:1 present. If the authors cannot delineate such isomers, then they should revert to another level of lipid annotation. In this case, the authors should use species-level annotation PC 36:2 instead of PC 18:0-18:2 (which in this case is misleading). This issue also pertains to especially TG species. It cannot be that there are no FA isomers in plasma and serum. If the authors do not have a routine to resolve and quantify such isomers, and only rely on survey TOF MS data for quantification, then species-level annotation should be used and not molecular species-level.

-On a related note, please incorporate a supplementary figure with extracted ion chromatograms, CCS profiles and annotated TOF MS/MS spectra for the isomers TG 18:1_18:1_20:4 and TG 18:1_20:4_18:1, TG 16:0_18:1_20:4 and TG 18:1_18:2_18:2 as well as TG 18:2_20:4_22:6 and TG 20:4_20:4_20:4. Both for genuine plasma samples as well as 1:1 mixture of synthetic standards. This data can probably illustrate very nicely the utility of the ion mobility separation; or again help underpin that the revolving power primarily stem from the chromatography. Very interesting!

-Please provide a supplementary figure with extracted ion chromatograms for Cer d18:1_16:0 as [M+H]⁺ as well as [M+H-H₂O]⁺. It is well-known that ceramides can undergo in-source fragmentation and that this is instrument dependent. Similarly, also PC (as well as SM and LPC) species can undergo in-source fragmentation in negative ion mode to yield a demethylated analog (that is isomeric with endogenous dimethyl-phosphatidylethanolamine). Thus, please provide extracted ion chromatograms for PC 34:2 as a formate adduct (i.e., m/z 802.5604) and its demethylated counterpart (i.e., m/z 742.5392). Overall, these important analytical details should not be ignored.

-Supplementary Data 8 Analyte List: Please provide a detailed structural characterization of PC O-24:2/20:4 that shows extracted ion chromatograms, ion mobility profiles and annotated TOF-MS/MS spectra for positive and negative ion mode data. This should be accompanied by chemical reaction

schemes that shows putative structures of detected fragment ions, their calculated (not measured!) m/z values as well as the (average) ppm error for every fragment.

-Supplementary Data 8 Analyte List: Please incorporate extracted ion chromatograms for the protonated precursor of the uncommon lipid species SM 34:1;3O as well as a putative precursor with a m/z value of 1.00335 amu less (i.e., a possible co-eluting isotope interference). Show also ion mobility profiles and an annotated survey TOF-MS spectrum with a m/z range of ± 5 amu around the m/z value of protonated SM 34:1;3O. Basically, one wonders whether there is any type of isotope correction implemented to remove biases from co-eluting isotope interferences both in terms of identification and quantification. If not, this probably explains the difference in molar concentrations obtained using ISTD PC 17:0/14:1 versus ISTD PC 15:0/18:1-d7?

-Supplementary Data 8 Analyte List: The authors claim to baseline separate various sn-positional LPA, LPG and LPI isomers, but only denote the detection of sn-1 isomers. Please provide data/figure showing baseline separation of two or three pairs of sn-1 and sn-2 LPA, LPG and LPI positional isomers.

-Supplementary Data 8 Analyte List: How come there are three sn-positional isomers of LPC 16:0, of which two have identical CCS values and only marginally different RT values? Please illustrate the chromatographic and IMS baseline separation of the three sn-positional isomers.

-Supplementary Data 8 Analyte List: Please update the data file with a column reporting the calculated m/z values of precursor ions as well as m/z values of detected fragment ions. The latter can be inserted, for each lipid precursor (i.e., each line), as a string of m/z values with five decimals separated by the character "|" (similar to the supplementary data from Vasilopoulou et al. (2020)).

-Supplementary Data 8 Analyte List: It is strange that the main adduct ion of LPG species in positive ion mode is a proton whereas CCS values are primarily listed for ammoniated counterparts. All lipid molecules of a particular lipid class are expected to ionize with the same adduct ion(s). Thus, it is strange that some LPGs are listed with CCS values for ammoniated precursors whereas others are listed with protonated precursors. Please address this inconsistency.

-Please provide Supplementary Data files (positive and negative) with peak areas for all confident lipid annotations in all analyzed plasma samples (n= 64 plus blanks and calibrants). This file can feature columns with lipid name, adduct ion, measured m/z value (average, centroid and standard deviation), calculated m/z value and CCS value (average, centroid and standard deviation). From this, one can in principle easily calculate and fact-check the molar concentrations of the reported lipid

analytes. In addition, an additional sheet with estimated molar concentrations can also be provided (for people to benchmark their own calculations towards that of the authors).

-Please provide equations for how molar concentrations of lipid analytes in plasma and serum samples are computed as well as that of blood spots. These equations should feature variables that denote the volume of plasma or serum, the spike amount of internal standard as well as variables used for 'multi-point quantification' and 'one-point quantification'. The authors should also provide a Supplementary Data file with molar amounts of spiked standards.

-Supplementary Data 3: Sheet pos quan data: correct name of header in column D to "TOF pos". Add standard deviation values to sheets neg quan data and pos quan data. Add adduct information to sheets neg descriptors and pos descriptors.

Reviewer #2 (Remarks to the Author):

The topic of this manuscript is very exciting because it integrates the analytical strengths of TIMS-TOF with a concept of automated lipid identification and also quantification. The identification part is generally done very thoroughly with only some minor comments / additions necessary. Firstly it would be good to have not only RT, m/z and CCS values of all the identified lipids in the supplementary Excel files, but also the MS/MS fragments detected alongside. Furthermore it would be interesting to see chromatograms for PA and LPA because these lipid classes are due to their primary phosphate group often 'troublemakers' in reversed phase chromatography and tend to result in rather smearing peaks. When it comes to FAHFAs it is important to ensure that it is really these lipids which are detected and not just FA dimers. Was this issue taken care of? The pitfalls surrounding the analysis of FAHFAs are well documented in a recent publication (Alisa B. Nelson et al., JLR, 2022).

Although the quantitative lipid values from several technical approaches fit together quite well – which indicates that the overall quantitative performance of the method is acceptable – there are still two remaining issues concerning quantification. First, the quantification used is not level 2 but for most lipids rather level 3, because for level 2 the analyte and the IS would have to co-elute. Given the fact that reversed phase HPLC was used, this seems to be highly unrealistic if just one or even a couple of IS per lipid class were used. This would e.g. be possible with a HILIC separation. Furthermore it is not clear from the manuscript if the calibration curves (multiple point) were performed in the samples matrix (plasma, serum) or if they were just performed in an organic solvent. Please clarify.

Minor point:

In line 597 the term 'isobaric' should be substituted by 'isomeric', because this seems to be what the authors want to express.

Reviewer #3 (Remarks to the Author):

Overall:

Lerner et al. make a valiant effort to introduce a quantitative and reproducible workflow for clinical lipidomics. This is very important and has not been accomplished in a comprehensive fashion. Generally, I think they need more evidence to prove they succeeded:

1) Reproducible (LIKELY SUCCEEDED) – I think they give ample evidence that their method are reproducible over a 2 week period, but a comparison across multiple extractions using their automated approach on different days would be useful. But I think this portion is good.

2) High throughput (LIKELY SUCCEEDED) – it looks like their automated extraction does make this high throughput, although data-processing time should be mentioned. Excellent work on automation! (although more validation or at least reporting of the results comparing manual and automated extraction is needed).

3) Quantitative (LIKELY FAILED) – Lipidomics using reverse phase chromatography is not quantitative by design, this has been discussed in multiple manuscripts and is agreed upon by the lipidomics community. THE AUTHORS SHOULD NOT CLAIM THEIR METHOD IS QUANTITATIVE WITHOUT PROOF, rather they should state semi-quantitative or relative quantitation throughout the manuscript. While they compare their method with many other non-quantitative results from other labs and methods including MRM, this is insufficient. They must spike in a large number of endogenous lipids which can be purchased into a stripped matrix such as serum, and show they get values within a desirable range.

4) Annotation accuracy of near 100% (NOT ENOUGH EVIDENCE) – a table with all results for datasets including fragments observed, retention times and delta, CCS values and delta, exact mass and delta... would be helpful to determine your confidence. Supplementary information shows correct annotation of some strange lipids, but this is only a very few examples. Too many times have we determined an extremely high false positive rate in lipidomics, so all evidence must be included (if already included, sorry please direct me).

In conclusion I do not think the authors succeed in establishing what they set out to establish, specifically on absolute quantitation. I do not think this is of quality for Nature Methods, especially given Hiroshi et al. <https://www.nature.com/articles/s41587-020-0531-2> A lipidome atlas in MS-DIAL 4 has a manuscript which covers most of these aspects in a more comprehensive fashion

embedded into one of the most widely used lipidomics software. I do think it should be published elsewhere and is of value to the lipidomics community, and applaud this thorough work.

Minor:

“lipids other than traditional clinical parameters such as cholesterol, TGs, and 61 lipoproteins have been largely overlooked in the past in clinical research and applications” awkwardly phrased, rephrase.

“feature recursiveness in replicate extracts of reference plasma” what does this mean? Clarify and define.

“The capability of the here developed 4D-PASEF lipidomics platform...” Throughout the language structure and grammar is odd. Be reductionist, remove any extra phrases which are not necessary, this sentence is a run-on and wordy.

“ISTDs” make sure to define acronyms with first use

11,11% to 11.11%...

For your supplemental there are other fragments you can annotate as well:

RCO 10:0 RCO 14:0 RCO 16:0

155.143 211.2056 239.2369

Major:

“These developments were proved...” awkwardly phrased, rephrase... “to enable high-resolution separation of lipid molecules and distinction of otherwise unresolved isomers, such as those with different double bond positions or geometries.” Clarify what otherwise unresolved isomers signifies, I assume you mean with versus without ion mobility. In which case I would state this in less exaggerating language. Yes, certain isomers which are not separated in LC can be separated in ion mobility, but I generally find this the exception not the rule. LC has much higher resolving power, and maybe you can state some important cases where it matters biologically but do not oversell! Make sure the reader knows that most Sn1 and Sn2 isomers, most double bond positional isomers,

etc. will not be further resolved completely using ion mobility when not resolved by a long LC run. You can maybe learn more on the rapid analysis aspect being able to shorten LC runs, but even with ion mobility you are giving up a lot.

“...we reduced the overall processing time for 96 samples to 3 h...” awesome! I have not seen many successful application of robotics for lipid extraction, so this is very exciting, make sure to include everything and any software files etc. in the supplemental so people can reproduce your method exactly!

Figure 1a: Not very informative and hard to follow. I would love to see RSDs with manual versus automated extraction, recoveries for all three methods (nor extraction, manual extraction, and automated extraction (which should be known amount spike in and concentration measure not a comparison of peak areas), and matrix effect defined better and for all comparisons. Also this is a table not a figure.

Figure 1c: put the name of the lipid / structure in the figure for quick reference. Also this goes to further my point that generally RT is better at separating isomers than mobility, mobility is not an orthogonal approach although in some cases it does give new information.

I would not cite (ACTUALLY DO NOT) (Vasilopoulou et al., 2020), see the following manuscript: “Quality control requirements for the correct annotation of lipidomics data | Nature Communications” which is a response to their work. After review of their data the authors found that THE MAJORITY of their annotations were VERY LIKELY false positives! If anything you can cite it and the response to show how your method overcomes their challenges.

Fig 2b, and c, where these for different extractions? This is necessary to show your robust automated extraction method. Each assay should be a different extraction, although comparing different runs across weeks is also valuable. Nice on the retention time reproducibility! This is very challenging, you should note different columns, if not ordered from the same batch, and eventually overtime you likely wont have this retention time reproducibility.

“However, the annotation requires additionally a generation of RT calibration data which increases the overall workflow” how else are you going to do identification with retention time? Eventually your retention times will change in a new experiment, new LC column batch, when adding a guard or as the column gets old... especially between labs, etc.

For your adducts MG, DG and CE should also have predominant NH₄ adducts... and also ether-linked and oxidized version of all the ones mentioned. Although maybe you did not detect these?

“Using only m/z- and spectral matching- based annotation afforded by MS Dial led to misassignment of several features to LNAPE instead of PE (Fig. 4a).” I believe there is the option to use retention time in MS-DIAL when you use the same methods, so this statement is not correct.

For lyso SN1/SN2 isomers for LPC fragmentation can be used to distinguish species...

“Finally, out of 15899 features initially detected from NIST SRM plasma in negative ion mode, highly reproducible stable unknown features, exhibiting a plasma dilution response, were identified...” very cool! Interesting method for determining unknown biological molecules.

“which gave rise to substantial variation in the quantification results compared to the inter-laboratory study by Bowden et al” I would take the results from Bowden et al.s study in terms of comparing values with a grain of salt. This study was well designed, but the results showed huge variations in quantified levels depending on which labs reported values. In other words, while there were consensus values reported, there was no consensus between labs, and so there is no way of knowing (also based on the study design) that these were correct consensus values.

This goes to another point: never use the word “quantitation” in lipidomics unless you are targeting a lipid or lipids with internal standards of those species (e.g. isotopically labeled) and have a calibration curve for each species. In HILIC you may be able to not have exact labeled standards. This is not a fault of your methods, but a general lack in good methods available for this work based on fundamental issues such as ion suppression, which is specific to retention time, etc. See: Software tool for internal standard based normalization of lipids, and effect of data-processing strategies on resulting values | BMC Bioinformatics. I see you have multi-point calibrations, and comparison to other studies and MRM, which is great. But 1) based on Figure 7 there is often a 2-fold change or more difference between studies, which is actually excellent for this work but I would not call this quantitative. 2) the comparisons to other studies is an issue since their work and consensus values were not based of quantified methods or validated in absolute terms... If you think your work is quantitative, then please have a spiked in lipids stripped matrix (you can purchase hundreds of endogenous lipids), or QC sample with absolute values to compare your values against. This is not easy but needed for this claim.

See (although I have not tried this method): Preparation of Lipid-Stripped Serum for the Study of Lipid Metabolism in Cell Culture – Bio Protoc.

Also see the conclusions from the following manuscript: “Quantification of Lipids: Model, Reality, and Compromise - Thus, absolute quantification for complex lipids should not be established with mass spectrometry detection, we should only propose data in relative quantification”

You mention high sensitivity of the PASEF platform, can you benchmark this against more traditional methods? I would assume sensitivity decreases with ion mobility based on fundamentals... although signal to noise may increase... please have a comparison table (maybe in the supplemental).

REVIEWER COMMENTS

Reviewer #1 (Remarks to the Author):

- The manuscript by Lerner et al., provides a nice insight into the utility of PASEF technology for clinical lipidomics. The manuscript nicely outlines various steps undertaken to validate and benchmark the technology for such analyses. Furthermore, the quality of the lipidomics data appears to be better than that of Vasilopoulou et al. (2020); which is very good and seems to underscore that also PASEF technology can be used for screening purposes in clinical lipidomics.

Answer: We thank the reviewer for the positive appreciation.

- Nevertheless, the manuscript and the work have a range of issues that can be addressed. First, the overall readability of the manuscript can be improved. As it stands, there is a lot of effort devoted to using buzzwords rather than articulating more concisely and exactly how the results were obtained. Notably, the authors somewhat inflate the added value of the TIMS dimension in lipid annotation and quantification. Moreover, from reading the main text it is not clear whether PASEF data (i.e., TIMS-TOF MS/MS data) is used for quantification or whether survey TOF MS data is used for quantification. Here, the latter seems to be the case albeit the authors use the then inappropriate term "4D-PASEF quantification".

Answer: Thank you for pointing this out. Indeed, we have used the survey TOF MS data for quantification and adjusted accordingly in the manuscript the terms. We have thoroughly revised the manuscript to improve readability and applied the corrections and suggestions of the reviewer also from the edited pdf document for which we are grateful. It was beyond our intention to inflate the value of TIMS or to market Bruker, hence we addressed this by revising thoroughly the main text.

- Second, it would be outstanding if the authors had devoted more effort to assess the analytical sensitivity and dynamic quantification range of the platform using pairs of lipid class-specific, stable isotope-labelled standards spiked into plasma samples, follow by lipid extraction and MS analysis. This should be done for the majority of lipid classes already assessed by the multi-point strategy (Fig. 2a) as well as cholesterol, CE, TG and DG lipids. Plus, it should be done using plasma matrices with hyperlipidemia and normolipidemia. Dynamic quantification range curves (based on non-smoothed not regression-corrected data!) can be presented as supplementary information; and LOD and LOQ values can be added next to Fig. 2a and 2b. Moreover, LOD and LOQ values should be compared to that obtained by the LC-MRM analysis. The point here is, that the authors insinuate over and over

again that the 4D-lipidomics platform is very sensitive, but provide no data to support this claim. I believe that the sensitivity of the 4D-lipidomics platform is comparable to that of most other lipidomic platforms, including those based on standard QqTOF and Orbitrap technology. Thus, the authors might eventually want to downplay the hype about the high sensitivity of the timsTOF technology.

Answer: *We have followed the reviewer's suggestion and reworded the text to downplay the sensitivity of the timsTOF. It was beyond the scope of the paper to compare this platform with other lipidomics platforms. We based the claim of sensitivity strictly on the higher number of lipids that we report with molar concentrations determined by our platform compared to the inter-laboratory reports. In that sense, we referred to the overall sensitivity of the workflow. We do share the reviewer's opinion that TimsTOF has comparable sensitivity to other untargeted lipidomic platforms; unfortunately, we do not have Orbitrap in the lab to compare the workflow between the platforms and lipid qualification strategy outlined here. The LOD and LOQ for targeted lipid analysis based on MRM have been previously evaluated by us using mouse plasma and a citation of this paper has been included for reference, and it will be surely interesting in the future to see whether targeted assay using PRM on TIMS instrument has comparable results to MRM.*

Concerning the use of stable isotope-labeled standards spiked in the plasma samples and especially normo and hyperlipidemia plasma. We did now include in the manuscript extraction and quantification of plasma using class-specific deuterated standards (many of the deuterated became available only during the review process). This was done for 64 plasma samples extracted over two-plates: we provided the data (including the peak areas of analytes and standards) in (Supplementary Data 6), and assessed the inter-plate assay using a set of deuterated standards (Figure 5b, Supplementary Data 7). We also provided a comparison between the first set of standards used and the new deuterated ones in terms of obtained molar concentrations and pointed out the influence and choice of internal standards on the lipid molar concentrations (Supplementary Data 7)- which is pointed out by the lipidomic community. Recovery and matrix effects were also included in Table 1 for this set of standards. The data on regression parameters were added in Supplementary Data 12. The data were acquired with a smooth on function as per factory settings- so non-smothered data are therefore not available.

We were, unfortunately, unable to procure neither from commercial sources nor local hospitals reference samples of normo- and hyperlipidemia - both sources conveyed to us that due to the Corona situation they either did not have in stock or could not collect in a reasonably foreseeable future. Therefore, we performed these new experiments with NIST plasma. Of note, we selected lipid features in Figure 3 based on plasma dilution response (0.0625 μ L to 2 μ L plasma on the column) to address in part the issue of plasma matrix.

We do agree with the reviewer that using normo- and hyperlipidemia samples would be outstanding and we think that a systematic study of the performance of all currently available internal standards/calibrants also in certified lipid-stripped matrix and their influence on quantification output would be a necessary prospective study. We pointed out this in the manuscript as well.

- Third, the overall efficacy of the 4D-lipidomics platform is highlighted by the ability of the platform to accurately discriminate whether analyzed blood samples were obtained as plasma, serum, whole blood, venous DBS and finger-prick DBS. Here, it would have been more interesting, and relevant, had the utility of the platform been showcased by analyzing a limited cohort of, for example, ~50 subjects with hyperlipidemia and normolipidemia.

Answer: *Indeed, we agree it would be relevant to use normo- and hyperlipidemia. We do, however, consider that the ability to discriminate the blood matrices phenotypes does highlight the platform's efficacy considering that we have analyzed here 180 samples of four individuals, that plasma, serum, whole blood, venous DBS, and finger-prick DB exhibit different lipidome/matrix, and MRM was used for 45 samples to cross-validate.*

- Finally, I have attached an annotated pdf file for the authors to try to use to improve their manuscript. This pdf file features general comments and suggestions, emphasized by a red vertical line on the right.

Answer: *we have thoroughly applied the correction and addressed the suggestions reviewer asked.*

General comment on tims-separation, added values of CCS: *Certainly, chromatographic resolution is superior to mobility resolution. As we focused on developing a high-throughput platform, the routine, automatic confident annotation is essential for the processing of large-scale clinical data. In this context, added quality data attribute such as CCS is instrumental for feature selection (Figure 3) and additional annotation confidence criterion. Hence the CCS values were used as an additional data attribute for confident annotation and confirmation thereof, curation of multiple (see Supplementary Data 1 for examples of CCS-based data curation). We made this point in the Result and Discussion section (marked in the text), introduced examples of mobilograms that can be used for the detection/confirmation of isomers as well as outlined in the text a potential value of CCS for annotation independent of the chromatography method.*

We discussed FDP and provided examples of total annotation hits and confirmed hits obtained with and without CCS. For reviewer information: It is not possible in Metaboscape to extract data acquired with the ion mobility with a 3D algorithm, but only with 4D algorithm where CCS deviation is included as a parameter for feature extraction. Therefore, the evaluation of annotation with and without CCS was done only by removing CCS from the ClinLip Analyte list.

General comment on benchmarking to CER score and LDL: *We did calculate the CERT2 score based on data obtained from 96 NIST extracts (with data sets obtained one year apart basically), which resulted in consistent results for all extracts. In addition, we calculated the scores for two individuals from the Vampire study, over the three-time points, which equally rendered consistent results for all the time points for both individuals. We included and discussed this in the Results section and the data are included in Supplementary Data 8. We did not have reference data on LDL content from NIST plasma or individuals. The cholesterol values were only referenced to the inter-laboratory study.*

Comment on manual vs automatic extraction: *We included more data on manual versus automatic extraction, i.e. for additional deuterated standards in Table 1. We have stated in the manuscript that the data are similar, and we removed the subjective statement about error-prone manual handling. Indeed, we do not have a conflict of interest here. For reviewer information only: We have extensive experience with routine manual preparation of dozens of samples a day: both with skilled technicians and we were lucky to have also talented one-year PhD. From this experience, this work is laborious and tedious to maintain a high level of standardization over large sample cohorts, particularly time-window for complete sample preprocessing and preparations across all samples. For these reasons and for obvious throughput we would, of course, prefer to have automatic extraction that works as well.*

Comment on diurnal oscillations of lipids: *We absolutely agree with the reviewer that this is an essential aspect to be considered for lipids. We made this point in the results and discussion and emphasized the need for appropriate sampling and general study design to mitigate this. We think that the predictive power of lipids, in addition to the traditional ones, has been proven by newly introduced CERT scores, and also emphasized by the study cited in the discussion where lipidomics and genetic data were evaluated. Indeed, such developments can only be done with appropriate SOPs for sampling and processing.*

Random forest classification: *was done in the study for the 180 samples of the Vampire study, which is suitable to distinguish the biological matrices based on the large number of input features (lipid data per sample). We introduced in Methods section and results additional explanation of the test's utility.*

Annotation score: *was defined in the materials and methods and in the text. It is compiled of the data attributes used for annotation quality in the Metaboscape software.*

The influence of the position on the plate on the quantified data: *was included in Supplementary Figure 1 and discussed in the manuscript for CE and TG species.*

Fig 5 Average Inter-day CV should be lower than 5 %: *it would be indeed ideal. From our experience with MRM and literature survey, such excellent CVs are primarily obtained with the MRM method. For untargeted lipidomics, we have seen similar CV values in the literature. As we added 64 extracts prepared with deuterated standards for which inter-plate assay was evaluated (Figure 5b, and Supplementary Data 7) similar values were obtained for the new data.*

Number of Features in Figure 3: *15899 features are obtained with no filter and with a threshold of 100 counts for peak detection and 200 counts for feature extraction. As stated in Figure 3 this number dropped to 3013 when the first filter parameter was applied. Based on the individual analysis of the 200 standards in-source fragmentation was not a common occurrence, see below for ceramide and demethylation of PC.*

Comment on cholesteryl ester quantification: *We included a supplementary (Supplementary Figure 8) to highlight the discrepancy between CE and cholesterol internal standard response in replicate analysis. We agree with the reviewer that a deuterated CE (which unfortunately was not available at lipidmaps) would be more appropriate and as per suggestion will do so in the future.*

Comment on C1P and S1P in Table of standards: *The Table was for the QC samples where we used these species. CerP was detected in one of Vampire participants and quantified with Ceramide internal standard.*

- Other important comments (not exhaustive):

It is awkward, and almost mysterious, why the authors do not detect any FA chain isomers (e.g., Supplementary Data 8 Analyte List). For example, it cannot be that the authors only detect PC 18:0-18:2 in human plasma. There must be some PC 18:1-18:1 present. If the authors cannot delineate such isomers, then they should revert to another level of lipid annotation. In this case, the authors should use species-level annotation PC 36:2 instead of PC 18:0-18:2 (which in this case is misleading). This issue also pertains to especially TG species. It cannot be that there are no FA isomers in plasma and serum. If the authors do not have a routine to resolve and quantify such isomers, and only rely on

survey TOF MS data for quantification, then species-level annotation should be used and not molecular species-level.

Answer: Indeed, only survey TOF MS was used for quantification. Certainly, it will be valuable to develop in the future fragmentation-based quantification which would allow more individual isomers quantification. We pointed out in Materials and methods that the most abundant isomer was listed in (Supplementary Data 6)(now including Supplementary Data 7), but that other isobars are present. This was also illustrated by the TG fragmentation example as well as Supplementary Figure 13d. The 200 standards we used to create the library had defined structures and therefore we aimed to annotate/delineate these species were present in plasma from other possible isomers of a given species-level composition. The same is true for the library curated in plasma. This Analyte list should permit the reader to infer RT and CCS references for defined structures.

PC 18:1 -18:1 overlap (as observed in negative ion mode) with an additional isomer of PC 18:0-18:2 (other than the one we have a standard for) which shows a higher abundance than PC 18:1 -18:1 (Figure below). For the additional isomer of PC 18:0-18:2 we do not have RT, CCS reference from standard or additional MS/MS information to annotate its structure; therefore it was manually excluded from the high confident list (Supplementary Data 6). Indeed, this is another example of utility of extending standard library and of fragment-based quantification.

Figure (For Reviewer Information only):

- On a related note, please incorporate a supplementary figure with extracted ion chromatograms, CCS profiles and annotated TOF MS/MS spectra for the isomers TG 18:1_18:1_20:4 and TG 18:1_20:4_18:1, TG 16:0_18:1_20:4 and TG 18:1_18:2_18:2 as well as TG 18:2_20:4_22:6 and TG 20:4_20:4_20:4. Both for genuine plasma samples as well as 1:1 mixture of synthetic standards. This data can probably illustrate very nicely the utility of the ion mobility separation; or again help underpin that the revolving power primarily stem from the chromatography. Very interesting!

Answer: we provided these data in Supplementary Material (Supplementary Figure 5) and discussed in the text the results. TG 18:2_20:4_22:6 and TG 20:4_20:4_20:4 were not available as synthetic standards and their chromatogram/mobilogram were extracted from plasma only. Yes, these data confirm that chromatography is still superior to mobility separation. We included also an ultra-high mobility resolution to showcase how targeted high mobility resolution can confirm/underpin isomer presence for short chromatography gradients used to higher throughput.

- Please provide a supplementary figure with extracted ion chromatograms for Cer d18:1_16:0 as [M+H]⁺ as well as [M+H-H₂O]⁺. It is well-known that ceramides can undergo insource fragmentation and that this is instrument dependent. Similarly, also PC (as well as SM and LPC) species can undergo insource fragmentation in negative ion mode to yield a demethylated analog (that is isomeric with endogenous dimethyl-phosphatidylethanolamine). Thus, please provide extracted ion chromatograms for PC 34:2 as a formate adduct (i.e., m/z 802.5604) and its demethylated counterpart (i.e., m/z 742.5392). Overall, these important analytical details should not be ignored.

Answer: We did provided EIC for these species and discussed in the main text. For ceramide in positive we do observe loss of water, but no demethylation for PC.

- Supplementary Data 8 Analyte List: Please provide a detailed structural characterization of PC O-24:2/20:4 that shows extracted ion chromatograms, ion mobility profiles and annotated TOF-MS/MS spectra for positive and negative ion mode data. This should be accompanied by chemical reaction schemes that shows putative structures of detected fragment ions, their calculated (not measured!) m/z values as well as the (average) ppm error for every fragment.

Answer: We did include data for this species in Supplementary Figure 13a.

- Supplementary Data 8 Analyte List: Please incorporate extracted ion chromatograms for the protonated precursor of the uncommon lipid species SM 34:1;3O as well as a putative precursor with a m/z value of 1.00335 amu less (i.e., a possible co-eluting isotope interference). Show also ion mobility profiles and an annotated survey TOF-MS spectrum with a m/z range of ± 5 amu around the m/z value of protonated SM 34:1;3O. Basically, one wonders whether there is any type of isotope correction implemented to remove biases from co-eluting isotope interferences both in terms of identification and quantification. If not, this probably explains the difference in molar concentrations obtained using ISTD PC 17:0/14:1 versus ISTD PC 15:0/18:1-d7?

Answer: We included data for this species in the survey TOF-MS spectrum with a m/z range of ± 5 amu around the m/z value of protonated SM 34:1;3O. We do not see interference from a m/z value of 1.00335 amu less Supplementary Figure 7c. On a general note, the choice and performance of internal standards have been recognized by the lipidomic community to influence quantification output, and is supported by our now extended data, and the current initiatives to harmonize lipidomic protocols in this context are essential. In line with this, a prospective systematic evaluation of all standards' performance for the Tims platform will be necessary (as mentioned also in the manuscript).

- Supplementary Data 8 Analyte List: The authors claim to baseline separate various sn-positional LPA, LPG and LPI isomers, but only denote the detection of sn-1 isomers. Please provide data/figure showing baseline separation of two or three pairs of sn-1 and sn-2 LPA, LPG and LPI positional isomers.

Answer: We included chromatograms and mobilograms for these species and also discussed in the text the potential of mobilogram-based confirmation of overlapping isomers (Supplementary Figure 4). In plasma, only the detected sn isomers were denoted Supplementary data 3. The RT, and CCS for the sn1 and sn2 isomers inferred from standards are listed in Supplementary data 1.

- Supplementary Data 8 Analyte List: How come there are three sn-positional isomers of LPC 16:0, of which two have identical CCS values and only marginally different RT values? Please illustrate the chromatographic and IMS baseline separation of the three sn-positional isomers.

Answer: This was a double entry error and was corrected. Thank you for pointing that out.

- Supplementary Data 8 Analyte List: Please update the data file with a column reporting the calculated m/z values of precursor ions as well as m/z values of detected fragment ions. The latter can be inserted,

for each lipid precursor (i.e., each line), as a string of m/z values with five decimals separated by the character “|” (similar to the supplementary data from Vasilopoulou et al. (2020)).

Answer: *We updated the data file similar to Vasilopoulou et al. (2020) Supplementary data 6.*

- Supplementary Data 8 Analyte List: It is strange that the main adduct ion of LPG species in positive ion mode is a proton whereas CCS values are primarily listed for ammoniated counterparts. All lipid molecules of a particular lipid class are expected to ionize with the same adduct ion(s). Thus, it is strange that some LPGs are listed with CCS values for ammoniated precursors whereas others are listed with protonated precursors. Please address this inconsistency.

Answer: *We corrected this- the entries were pasted in the wrong column. Thank you for pointing out this error in Supplementary data 1 and 3.*

- Please provide Supplementary Data files (positive and negative) with peak areas for all confident lipid annotations in all analyzed plasma samples (n= 64 plus blanks and calibrants). This file can feature columns with lipid name, adduct ion, measured m/z value (average, centroid and standard deviation), calculated m/z value and CCS value (average, centroid and standard deviation). From this, one can in principle easily calculate and fact-check the molar concentrations of the reported lipid analytes. In addition, an additional sheet with estimated molar concentrations can also be provided (for people to benchmark their own calculations towards that of the authors).

Answer: *We provided these data in Supplementary data 6. See also peak areas and 4D descriptors in Supplementary data 7.*

The list of external and internal standards was updated with spiked concentrations of ISTDs and concentration range for external standards.

- Please provide equations for how molar concentrations of lipid analytes in plasma and serum samples are computed as well as that of blood spots. These equations should feature variables that denote the volume of plasma or serum, the spike amount of internal standard as well as variables used for ‘multi-point quantification’ and ‘one-point quantification’. The authors should also provide a Supplementary Data file with molar amounts of spiked standards.

Answer: *We included the equations in Materials and Methods and included the molar amounts of spiked standards also for the new set of standards used in Supplementary Figure 6.*

-Supplementary Data 3: Sheet pos quan data: correct name of header in column D to "TOF pos". Add standard deviation values to sheets neg quan data and pos quan data. Add adduct information to sheets neg descriptors and pos descriptors.

The sheets were accordingly corrected and updated (Supplementary data 6 and 7) Thank you for pointing that out.

Reviewer #2 (Remarks to the Author):

- The topic of this manuscript is very exciting because it integrates the analytical strengths of TIMS-TOF with a concept of automated lipid identification and also quantification. The identification part is generally done very thoroughly with only some minor comments / additions necessary.

Answer: *We thank the reviewer for the appreciative words.*

- Firstly, it would be good to have not only RT, m/z and CCS values including deviation thereof, of all the identified lipids in the supplementary Excel files, but also the MS/MS fragments detected alongside.

Answer: *We have added also the MS/MS fragments and their annotation Supplementary data 6*

- Furthermore it would be interesting to see chromatograms for PA and LPA because these lipid classes are due to their primary phosphate group often 'troublemakers' in reversed phase chromatography and tend to result in rather smearing peaks.

Answer: *Yes, indeed they are troublemakers and unfortunately remained so. We have included in Supplementary Figure 4b EIC of lysospecies, where also the LPA tailing is visible. Unfortunately, we cannot overcome this by chromatography.*

- When it comes to FAHFAs it is important to ensure that it is really these lipids which are detected and not just FA dimers. Was this issue taken care of? The pitfalls surrounding the analysis of FAHFAs are well documented in a recent publication (Alisa B. Nelson et al., JLR, 2022).

Answer: *Indeed, this is an important point, we thank the reviewer for pointing it out. In the first place we did have standards for FAHFAs and we inferred the RT and fragmentation from the standard's analysis, and secondly, we used the diagnostic ions pointed out in the paper (Alisa B. Nelson et al., JLR, 2022) for annotation. Example for MS/MS fragmentation is included in the Supplementary Figure 13c*

- Although the quantitative lipid values from several technical approaches fit together quite well – which indicates that the overall quantitative performance of the method is acceptable – there are still two remaining issues concerning quantification. First, the quantification used is not level 2 but for most lipids rather level 3, because for level 2 the analyte and the IS would have to co-elute. Given the fact that reversed phase HPLC was used, this seems to be highly unrealistic if just one or even a couple of IS per lipid class were used. This would e.g. be possible with a HILIC separation. Furthermore it is not clear from the manuscript if the calibration curves (multiple point) were performed in the samples matrix (plasma, serum) or if they were just performed in an organic solvent. Please clarify.

Answer: *We thank the reviewer for raising this issue. We corrected the level 2 to 3. Indeed, for reversed-phase chromatography, the use of one standard or even a couple is not ideal. However, it is also true that unfortunately the availability of standards to properly cover the chromatographic range is scarce and we consider this relative quantification. Our aim is to achieve a reproducible quantification with as much cross-referenced data as possible within the method's confinements, in order to be able to reproducibly identify a lipidome phenotype, and with a reasonable molecular resolution that can enhance phenotype differentiation, hence the choice of reversed-phase and IMS. We have now added in the manuscript a set of experiments for 64 plasma extracts using deuterated internal standards and compared the performance of the two sets of standards on the quantification output. These data are added in Supplementary data 7, Figure 5b, and the comparison is discussed in the main text. In line with reviewer's view and the lipidomic community's view on the choice and influence of standards on the quantification output we pointed out these aspects in the manuscript. Calibration curves were prepared in organic solvent, indeed. Unfortunately, certified, commercially available lipid-stripped matrix is not available; but hopefully such matrix will become available, especially useful for large-scale cohorts screening and for harmonization of protocols across laboratories.*

Minor point:

In line 597 the term 'isobaric' should be substituted by 'isomeric', because this seems to be what the authors want to express.

Answer: *We corrected this point.*

Reviewer #3 (Remarks to the Author):

Overall:

Lerner et al. make a valiant effort to introduce a quantitative and reproducible workflow for clinical lipidomics. This is very important and has not been accomplished in a comprehensive fashion. Generally, I think they need more evidence to prove they succeeded:

Answer: *We thank the reviewer for the encouraging words.*

1) Reproducible (LIKELY SUCCEEDED) – I think they give ample evidence that their method are reproducible over a 2 week period, but a comparison across multiple extractions using their automated approach on different days would be useful. But I think this portion is good.

Answer: *We included inter-plate extraction which was performed with a new set of deuterated internal standards and included the full data in Supplementary data 7. We included the inter-plate reproducibility in Figure 5b, and discussed the comparison between the standard's performance in the main text.*

2) High throughput (LIKELY SUCCEEDED) – it looks like their automated extraction does make this high throughput, although data-processing time should be mentioned. Excellent work on automation! (although more validation or at least reporting of the results comparing manual and automated extraction is needed).

Answer: *We included this comparison as a main figure, see Figure 1a and Table 1.*

3) Quantitative (LIKELY FAILED) – Lipidomics using reverse phase chromatography is not quantitative by design, this has been discussed in multiple manuscripts and is agreed upon by the lipidomics community. THE AUTHORS SHOULD NOT CLAIM THEIR METHOD IS QUANTITATIVE WITHOUT PROOF, rather they should state semi-quantitative or relative quantitation throughout the manuscript. While they compare their method with many other non-quantitative results from other labs and methods including MRM, this is insufficient. They must spike in a large number of endogenous lipids which can be purchased into a stripped matrix such as serum, and show they get values within a desirable range.

Answer: *We agree with the reviewer that this is relative quantification; we made sure to word accordingly in the manuscript. We aimed to assess and achieve a reproducible relative quantification so that a given lipidome phenotype can be reproducibly identified. In this sense, the consistent phenotype determined for the different blood matrices of the Vampire study showcases this ability of the platform. To further support the utility of the method, we also calculated CERT2 scores (per another reviewer's suggestion) from the NIST plasma extracts (n=96) and for samples from 2 individuals of the*

Vampire study, in order to assess if consistent scores are obtained, which were. We added this data to the manuscript and discussed the results.

We do consider that untargeted relative quantification if reproducible and cross-validated can be successfully used for marker discovery, as also demonstrated by new markers advanced in the clinical practice.

We certainly agree that for absolute quantification a different study design altogether has to be done and carried out, of course including lipid striped matrix with endogenous lipids, performance evaluation of internal standards, and application on reference normo- hyperlipidemia samples, etc. We are certainly interested in prospectively assessing absolute quantification using this instrument, and different chromatography and quantification strategies. This was beyond the purpose of this study.

4) Annotation accuracy of near 100% (NOT ENOUGH EVIDENCE) – a table with all results for datasets including fragments observed, retention times and delta, CCS values and delta, exact mass and delta... would be helpful to determine your confidence. Supplementary information shows correct annotation of some strange lipids, but this is only a very few examples. To many times have we determined an extremely high false positive rate in lipidomics, so all evidence must be included (if already included, sorry please direct me).

Answer: *We included all the fragment ions, RT, m/z, CCS and deviations thereof in Supplementary data 6. We have aimed at achieving high confidence annotation and hence select the lipids that can be routinely confidently annotated in large-scale studies. Therefore, we applied the stringent filtering criteria in Figure 3 and used procured and profiled 200 standards to infer accurate RT and CCS values.*

The raw data with fragmentation was and is available to the reviewer and reader as well.

- In conclusion I do not think the authors succeed in establishing what they set out to establish, specifically on absolute quantitation. I do not think this is of quality for Nature Methods, especially given Hiroshi et al. <https://www.nature.com/articles/s41587-020-0531-2> A lipidome atlas in MS-DIAL 4 has a manuscript which covers most of these aspects in a more comprehensive fashion embedded into one of the most widely used lipidomics software. I do think it should be published elsewhere and is of value to the lipidomics community, and applaud this thorough work.

Answer: *We are sorry if misleading with a lack of appropriate wording. Again, given that we use reverse-phase chromatography we are indeed referring to relative quantification, with the aim to be as reproducible as this analytical strategy permits.*

We are aware of the MS-DIAL manuscript and its value as software and embedded aspects. The purpose here is different from the one of the manuscript by Hiroshi et al. and it is tailored specifically to developing a high-throughput platform, automatic extraction with LC/Tims TOF for lipidomics that can be reliably used for marker discovery.

We hope that the reviewer finds merit in the revised manuscript for the lipidomic community and considers it suitable for Nat. Commun.

- Minor: “lipids other than traditional clinical parameters such as cholesterol, TGs, and 61 lipoproteins have been largely overlooked in the past in clinical research and applications” awkwardly phrased, rephrase.

Answer: *We rephrased this.*

- “feature recursiveness in replicate extracts of reference plasma” what does this mean? Clarify and define.

Answer: *We defined this in Materials and methods. It refers to 4D features that are detected in multiple samples in a batch (the minimum nr of samples in a batch for which 4D features have to be present are defined by the user in the processing software) and this criterion is used in the processing software to select features.*

- “The capability of the here developed 4D-PASEF lipidomics platform...” Throughout the language structure and grammar is odd. Be reductionist, remove any extra phrases which are not necessary, this sentence is a run-on and wordy.

Answer: *We thoroughly revised the text and we thank the reviewer for the input.*

-11,11% to 11.11%...

For your supplemental there are other fragments you can annotate as well:

RCO 10:0 RCO 14:0 RCO 16:0
155.143 211.2056 239.2369

Answer: *The assignments were included and “,” were exchanged.*

- Major: “These developments were proved...” awkwardly phrased, rephrase... “to enable high-resolution separation of lipid molecules and distinction of otherwise unresolved isomers, such as those with different double bond positions or geometries.” Clarify what otherwise unresolved isomers signifies, I assume you mean with versus without ion mobility. In which case I would state this in less exaggerating language. Yes, certain isomers which are not separated in LC can be separated in ion mobility, but I generally find this the exception not the rule. LC has much higher resolving power, and maybe you can state some important cases where it matters biologically but do not oversell! Make sure the reader knows that most Sn1 and Sn2 isomers, most double bond positional isomers, etc. will not be further resolved completely using ion mobility when not resolved by a long LC run. You can

maybe lean more on the rapid analysis aspect being able to shorten LC runs, but even with ion mobility you are giving up a lot.

Answer: *We have pointed out that LC has much higher resolving power and discussed the contribution of mobility separation primarily to isomer confirmation, multiple hits curation based on CCS values, and added data attributes for annotation confidence. We also included chromatograms and mobilograms of lysospecies and synthetic TG isomers to illustrate this point. Limitations of the method for separation are included in Discussion. We leaned more on the rapid analysis, as the reviewer suggested and aimed here. Certainly, even with ion mobility there is a lot of information that is missed with the current data processing settings. We do, however, hope that it sets out a good base for post-acquisition data mining for extended annotation and quantification, and we expect also that lipidomic community will include PRM developments for quantification of overlapped species.*

- "...we reduced the overall processing time for 96 samples to 3 h..." awesome! I have not seen many successful application of robotics for lipid extraction, so this is very exciting, make sure to include everything and any software files etc. in the supplemental so people can reproduce your method exactly!

Answer: *Thank you for the appreciation. The settings for the automatic lipid extraction are included in the materials and methods.*

- Figure 1a: Not very informative and hard to follow. I would love to see RSDs with manual versus automated extraction, recoveries for all three methods (nor extraction, manual extraction, and automated extraction (which should be known amount spike in and concentration measure not a comparison of peak areas), and matrix effect defined better and for all comparisons. Also this is a table not a figure.

Answer: *We exchanged the figure with Table 1 containing these data and updated the Figure 1a, for manual versus automated extraction.*

Figure 1c: put the name of the lipid / structure in the figure for quick reference. Also this goes to further my point that generally RT is better at separating isomers then mobility, mobility is not an orthogonal approach although in some cases it does give new information.

Answer: *We added the name of lipid. Yes, indeed it is clear from the figure that LC is superior to mobility and stated as such throughout the manuscript and discussion.*

- I would not cite (ACTUALLY DO NOT) (Vasilopoulou et al., 2020), see the following manuscript:

“Quality control requirements for the correct annotation of lipidomics data | Nature Communications” which is a response to their work. After review of their data the authors found that THE MAJORITY of their annotations were VERY LIKELY false positives! If anything you can cite it and the response to show how your method overcomes their challenges.

Answer: *Absolutely agree with the reviewer. We did cited the response paper to highlight this aspect.*

Vasilopolou paper was only cited in a neutral tone in the manuscript to refer to their CCS reproducibility.

- Fig 2b, and c, where these for different extractions? This is necessary to show your robust automated extraction method. Each assay should be a different extraction, although comparing different runs across weeks is also valuable. Nice on the retention time reproducibility! This is very challenging, you should note different columns, if not ordered from the same batch, and eventually overtime you likely wont have this retention time reproducibility.

Answer: *We added data on different plate extraction. We essentially have now 32 plasma extracts with remeasurement and inter-plate extraction (2 plates, n=32 per plate). For the sample number in this study we did not change the column. We do exchange columns, based on QC samples. For reviewer information only: we have acquired a large batch of samples since and we could run ca 1500-2000 samples per column). We do see overtime of course some deviation but based on QC we exchange column if RT deviation is outside the acceptable/calibration range.*

- “However, the annotation requires additionally a generation of RT calibration data which increases the overall workflow” how else are you going to do identification with retention time? Eventually your retention times will change in a new experiment, new LC column batch, when adding a guard or as the column gets old... especially between labs, etc.

Answer: *Yes, we rephrase that sentence as it was unclear.*

- For your adducts MG, DG and CE should also have predominant NH₄ adducts... and also ether-linked and oxidized version of all the ones mentioned. Although maybe you did not detect these?

Answer: *We do not detect the ether-linked or oxidized versions. Yes, the MG, DG, CE do have NH₄ adducts. Supplementary data 1,3, 6 and 7.*

- “Using only m/z- and spectral matching- based annotation afforded by MS Dial led to misassignment

of several features to LNAPE instead of PE (Fig. 4a).” I believe there is the option to use retention time in MS-DIAL when you use the same methods, so this statement is not correct.

Answer: *We rephrased this. The MS-DIAL library in the Metaboscape did not have an embedded option for including RT. We could only do that in an extra Analyte list which was linked to the spectral library.*

- For lyso SN1/SN2 isomers for LPC fragmentation can be used to distinguish species...

Answer: *We corrected this- they are not distinguishable by automatic spectral matching*

- “Finally, out of 15899 features initially detected from NIST SRM plasma in negative ion mode, highly reproducible stable unknown features, exhibiting a plasma dilution response, were identified...” very cool! Interesting method for determining unknown biological molecules.

Answer: *Thank you!*

- “which gave rise to substantial variation in the quantification results compared to the inter-laboratory study by Bowden et al” I would take the results from Bowden et al.s study in terms of comparing values with a grain of salt. This study was well designed, but the results showed huge variations in quantified levels depending on which labs reported values. In other words, while there were consensus values reported, there was no consensus between labs, and so there is no way of knowing (also based on the study design) that these were correct consensus values.

Answer: *Thank you for this. Indeed we take it with a grain of salt, therefore we also included reference to Wolrab. We used this as a reference/orientation not as an accurate benchmarking. We are curious and eager for the future initiative of lipidomic community to extend these studies and find consensus.*

- This goes to another point: never use the word “quantitation” in lipidomics unless you are targeting a lipid or lipids with internal standards of those species (e.g. isotopically labeled) and have a calibration curve for each species. In HILIC you may be able to not have exact labeled standards. This is not a fault of your methods, but a general lack in good methods available for this work based on fundamental issues such as ion suppression, which is specific to retention time, etc. See: Software tool for internal standard based normalization of lipids, and effect of data-processing strategies on resulting values | BMC Bioinformatics. I see you have multi-point calibrations, and comparison to other studies and MRM, which is great. But 1) based on Figure 7 there is often a 2-fold change or more difference between studies, which is actually excellent for this work but I would not call this quantitative.

Answer: *Thank you again for pointing this. In addition to the answer above to this issue we point out that our data emphasize as well that for reverse phase chromatography the choice of internal standards does make a difference in the outcome, more for some classes than for others based on the*

data, and that within standard set there is reproducible relative quantification which is good for phenotype ID, but between sets of standards different differences in reported values increase. We hope that it informs the reader about limitations of method as well and helps their design for future lipidomics with Tims instrument.

- the comparisons to other studies is an issue since their work and consensus values were not based of quantified methods or validated in absolute terms... If you think your work is quantitative, then please have a spiked in lipids stripped matrix (you can purchase hundreds of endogenous lipids), or QC sample with absolute values to compare your values against. This is not easy but needed for this claim. See (although I have not tried this method): Preparation of Lipid-Stripped Serum for the Study of Lipid Metabolism in Cell Culture – Bio Protoc. Also see the conclusions from the following manuscript: “Quantification of Lipids: Model, Reality, and Compromise - Thus, absolute quantification for complex lipids should not be established with mass spectrometry detection, we should only propose data in relative quantification”

You mention high sensitivity of the PASEF platform, can you benchmark this against more traditional methods? I would assume sensitivity decreases with ion mobility based on fundamentals... although signal to noise may increase... please have a comparison table (maybe in the supplemental).

Answer: *We referred to the sensitivity of the overall workflow based on reference to the nr of reported lipids by other studies. We did not compare sensitivity with or without ion mobility and with other methods in a systematic way, as this was outside the scope of the study. Ion usage metrics were published before and citations were provided in the manuscript.*

Therefore, we only included the LOD LOQ without comparing it to other methods. We did publish before our MRM method validation data on lipids, which we cited in the paper, but we used this for us as an internal reference only. We, unfortunately, do not have an instrument for untargeted lipidomics other than TIMS TOF. I think such a comparison should be done more systematically as the reviewer also pointed in lipid stripped matrices etc., and possibly compare MRM with PRM rather, and include systematic study with and without ion mobility on, altogether we think this would be a different study.

The overall workflow sensitivity was based on the reported lipids compared to inter-laboratory studies, and a high ion usage of PASEF which is basically also seen in the nr of features. Although for prospective routine confident annotation of large-scale data, we selected the recurrent and confident ones, for which we had appropriate 4D references, surely, there are more lipids that one could annotated when the recursive filters are decreased and expanded libraries are created.

REVIEWERS' COMMENTS

Reviewer #1 (Remarks to the Author):

The revised manuscript has to some extent been improved. At least now, the reader has a better chance to assess the metrics and limitations of the methodology and scrutinize some of the data, if they have the time. It is particularly interesting to be able to read, between the lines, that the TIMS dimension does not substantially improve the lipidomic data quality compared to other well-established platforms. Conversely, it is good to see that the TIMS-based platform has some promise for clinical lipidomics.

Nevertheless, the readability of the manuscript is still mediocre. For example, the authors spend too much text highlighting CV values instead of addressing other key performance metrics; especially the accuracy of the quantification (which is probably suboptimal given the authors compare such data using a log scale axis and heatmaps). It is also worrisome that the quantification of cholesterol, cholesteryl esters and triacylglycerols (and probably also diacylglycerol?) is not particularly good on the TIMS-based platform.

Other comments:

-the authors might want to change the title to “4D TIMS lipidomics for high-throughput clinical profiling of human blood samples”?

-“PASEF” is not abbreviated in the Abstract.

-Change the term “relative quantification” (back) to “quantification”. The authors are not doing relative quantification (according to the definition of the Lipidomics Standard Initiative; <https://lipidomicstandards.org/lipid-species-quantification/>). However, and as pointed out by Reviewer 3, the authors should make it clear to the reader that they are estimating molar abundances of lipids using “level-3 quantification”.

-TG is not abbreviated in the Introduction.

-Regarding false-positive identification using the MS Dial spectral library: Could it be, that the reason why searches with the ClinLip Analyte List does not pick up LNAPE species is because LNAPE (and NAPE) species are simply not on the ClinLip Analyte List?

-The term "ultra-ion mobility resolution" seems inadequate. Using this mode only marginally improves the peak width of ion mobilograms; which contrasts that of high resolution (~100,000) and ultra-high resolution (~10,000,000) in FTMS detection.

-The authors state "...likely due to hydrophobic precipitation with time...". Please provide a reference to scientific publications that have studied hydrophobic precipitation in the context of mass spectrometry-based lipidomics. This proposed hydrophobic precipitation could alternatively be due to contamination of the instrument? Which would be particular bad for high-throughput applications.

In relation to this, it is a bit strange that the authors do not include a figure panel where the 'precipitating' lipid analytes are normalized to the peak areas of lipid class-specific internal standards. Would this correct for the time-dependent "hydrophobic precipitation"?

Moreover, please add peak areas of endogenous cholesterol and PC 34:2 to Supplementary Figure 8a. Please add the peak area of endogenous PE 38:4 to Supplementary Figure 8b. It would be nice to see peak areas of lipid analytes that supposedly do not precipitate.

-The authors state "It is well known that diurnal changes of plasma lipidome, particularly of lipid mediators, readily occur". Please add one or two references to support the statement.

-In the Discussion the authors state "...resolution of isobaric phospholipids with small differences in fatty acyl chains...". Maybe the authors want to revise this to say: "...resolution of isobaric phospholipids with major differences in fatty acyl chains...".

-In the Discussion the authors state "... detailed description of genetic regulation of extended...". Maybe the authors want to revise this to say: "... detailed description of genetic association of extended...".

-The authors state "... 4D lipidomics platform, highly recommend it for future translation..." Please use another word instead of 'recommend'.

-Figure 2bi and 2bii. Please show EIMs for 1:1 mixtures of PC 18:1_18:1 Cis and Trans and PG 18:1_18:1 Cis and Trans; and not EIMs for the analysis of the individual standards.

-Supplementary Figure 13b and 13d. Please specify in the legend whether the spectra have been smoothed or if it is raw TOF MS/MS data?

-Regarding PC O-24:2/20:4: This seems to be a false-positive identification. To the best of my knowledge, the 'key' fragment with measured m/z 485.3383 cannot be generated by the applied fragmentation mechanism (or at least show this occurs for a synthetic PC O- standard). This ion, with odd-nominal mass, would probably imply a radical-containing fragment; which is unlikely. Please show corresponding data from positive ion mode analysis (i.e., XIC, EIM, survey TOF MS spectrum and full MS/MS spectrum, including a zoom from m/z 350-650 (there should be detection of a fragment ion with m/z 526.3292, diagnostic for the neutral loss of the ether-link chain).

BTW: the calculated m/z for the acyl anion of 20:4 should be m/z 303.2330 (i.e., 303.232954) and not m/z 303.2323 (-2.2 ppm off). Similarly, the calculated m/z for the demethylated precursor should be m/z 860.6539 (i.e., 860.653864) and not m/z 860.6533 (-0.7 ppm off). It seems to me that there is a calculation mistake for especially fragment m/z values in the ClinLip Analyte List? It appears as if the authors have forgotten to account for the mass of the electron in their calculations?

Reviewer #2 (Remarks to the Author):

No further comments.

Reviewer #3 (Remarks to the Author):

The authors generally addressed my comments well, and I appreciate the time and effort. Other than comments below their manuscript provides a methodology for semi-quantitative, high-coverage, reproducible, analysis of well annotated lipids for clinical studies. The proof will be in whether this method is widely adopted in the lipid community, which sadly often is not the case given most labs use separate methodologies. This is high-quality research with incremental

improvement on lipidomic methodologies, using many common practices. My final decision is that this should be published, but that it may not be revolutionary enough or novel enough for Nature Methods. I will leave that to the editor.

Major:

While they clarify the purpose of semi-quantification for observing relative changes and biomarker discovery in the response document (which I agree fully with their use case), the manuscript still uses quantification in some places. Make sure to always state relative quantification, especially in the abstract. For example you mention in the abstract: "We present ... encompassing automated extraction, confident

lipid annotation, and accurate quantification" so you are still misleading the reader... Relative quantitation, relative abundances, etc. should be used. Avoid concentration... etc.

Answer to the reviewers

REVIEWERS' COMMENTS

Reviewer #1 (Remarks to the Author):

- The revised manuscript has to some extent been improved. At least now, the reader has a better chance to assess the metrics and limitations of the methodology and scrutinize some of the data, if they have the time. It is particularly interesting to be able to read, between the lines, that the TIMS dimension does not substantially improve the lipidomic data quality compared to other well-established platforms. Conversely, it is good to see that the TIMS-based platform has some promise for clinical lipidomics.

Answer: We thank the reviewer for the appreciation.

- Nevertheless, the readability of the manuscript is still mediocre. For example, the authors spend too much text highlighting CV values instead of addressing other key performance metrics; especially the accuracy of the quantification (which is probably suboptimal given the authors compare such data using a log scale axis and heatmaps). It is also worrisome that the quantification of cholesterol, cholesteryl esters and triacylglycerols (and probably also diacylglycerol?) is not particularly good on the TIMS-based platform.

Answer: Indeed, we revised the text to improve the readability of the manuscript.

We have used CV values of individual lipids and average CV per lipid class to compare the performance of different sets of internal standards over 96 extracts, to compare targeted lipidomics using MRM with the 4D lipidomics method, and to compare our obtained lipid levels to the current consensus lipid values by Bowden et al, and by Wolrab et al.

We did not use heatmaps and log scales to compare data, but to strictly visualize/present in a readily readable fashion the comparison output for hundreds of lipids per dozens of replicate extracts. The graphical representation using heatmap also allowed the display of the broad concentration range of lipids in the biological matrix without compromising especially the display of upper and/or lower concentration ranges. The reader and reviewer can appreciate that we provided in the supplementary files individual lipid levels for each of the inter- and intra-assays and MRM- assays.

As per other key performance metrics concerning the accuracy of quantification. As we highlighted in the discussion too, the use of lipid-stripped matrices and plasma of normo- and hyperlipidemia plasma, along with new initiatives of lipidomic community to harmonize protocols and provide an updated consensus reference for accurate lipid levels in plasma will be instrumental to benchmark and reassess the performance metrics, and for us and the

interested reader to improve accordingly the quantification protocols. Nevertheless, as demonstrated also by the LBlood study, reproducible lipidome phenotyping is achievable by our method.

Cholesterol levels by 4D lipidomics have comparable values with the ones reported by Bowden et al. DGs exhibited an average CV of 18.4% for 64 extracts, and only a few consensus values for DGs are reported by Bowden et al., hence available for reference. TGs showed a CV of 19% intra-assay (n=32) and at one-year apart 27% CV (n=32). Hence, we consider the intra-day quantification assay as reproducible and reliable. We did, indeed, observed that CVs for TGs over 64 analyses (with samples residing in cooled autosampler for about 40 h) increases to 48,5 % and it is, in our opinion, important to have this evaluated and highlighted, especially since it can be missed in large scale cohort analysis. We and the reader can now design operational protocols to mitigate such changes over time, as we already stated in the manuscript.

We evaluated also now the CE quantification using level-3 TG ISTD, and the average CV for 32 extracts was found to be 18.4% and at one-year apart 26% (n=64), although the levels are substantially different than reported by inter-laboratory study, granted only a few CEs were in the study reported by Bowden et al. As discussed in the first revision, with the availability of deuterated CEs, quantification of CE levels with these standards can be assessed and implemented in routine profiling, and we are excited to adopt the new consensus CE levels as reference from the Lipidomics community reports.

Other comments:

- the authors might want to change the title to “4D TIMS lipidomics for high-throughput clinical profiling of human blood samples”?

Answer: We appreciate very much the suggestion and changed the title accordingly.

-“PASEF” is not abbreviated in the Abstract.

Answer: We adjusted the Abstract.

-Change the term “relative quantification” (back) to “quantification”. The authors are not doing relative quantification (according to the definition of the Lipidomics Standard Initiative; <https://lipidomicstandards.org/lipid-species-quantification/>). However, and as pointed out by

Reviewer 3, the authors should make it clear to the reader that they are estimating molar abundances of lipids using “level-3 quantification”.

Answer: We appreciate very much the suggestion and changed the term accordingly and provided the necessary citation in the manuscript.

-TG is not abbreviated in the Introduction.

Answer: We adjusted this.

-Regarding false-positive identification using the MS Dial spectral library: Could it be, that the reason why searches with the ClinLip Analyte List does not pick up LNAPE species is because LNAPE (and NAPE) species are simply not on the ClinLip Analyte List?

Answer: MS-Dial library does contain LNAPE spectra; however, during curation of the plasma library we did not find any spectra/feature in the plasma extract consistent with LNAPE. Therefore, LNAPE is not part of the ClinLip Analyte List. Even though MS Dial contains both PE and LNAPE and predicts different CCS values, the plasma PEs were still misannotated and we did not observe any differential mobilograms, i.e. CCS values, or chromatographic profiles consistent with distinct LNAPE and PE. It is though true that theoretically, one cannot exclude that LNAPEs are also present given that isobaric LNAPE and PE share certain fragment ions. To the best of our knowledge, so far LNAPEs were primarily reported in tissues, not in NIST human plasma SRM.

-The term “ultra-ion mobility resolution” seems inadequate. Using this mode only marginally improves the peak width of ion mobilograms; which contrasts that of high resolution (~100,000) and ultra-high resolution (~10,000,000) in FTMS detection.

Answer: We understand this concern for a 2x increase in the resolution of mobility. The term is defined as such by the instrument provider in the acquisition software.

-The authors state “...likely due to hydrophobic precipitation with time...”. Please provide a reference to scientific publications that have studied hydrophobic precipitation in the context of mass spectrometry-based lipidomics. This proposed hydrophobic precipitation could

alternatively be due to contamination of the instrument? Which would be particular bad for high-throughput applications.

In relation to this, it is a bit strange that the authors do not include a figure panel where the 'precipitating' lipid analytes are normalized to the peak areas of lipid class-specific internal standards. Would this correct for the time-dependent "hydrophobic precipitation"?

Moreover, please add peak areas of endogenous cholesterol and PC 34:2 to Supplementary Figure 8a. Please add the peak area of endogenous PE 38:4 to Supplementary Figure 8b. It would be nice to see peak areas of lipid analytes that supposedly do not precipitate.

Answer: We did provide citations for hydrophobic precipitation. Contamination is theoretically possible but we excluded it as a factor when regular cleaning and instruments performance checks are performed. The PC 34:2 and PE 38:4 are now included in Supplementary Figure 8b and they do not exhibit decay with time, nor is cholesterol which was included, attesting to the propensity of strong hydrophobic lipids like TGs and CEs to precipitate. The majority of high and medium abundance TGs are corrected by normalization to ISTD, i.e. quantification method (see the species with CVs less than 20% or 30%), and all the levels and CV values for all lipids are included in Supplementary Data 6 and 7. From our observation low abundant, and low abundant saturated TGs do not show a significant decay over time as opposed to the TG ISTD and they are conducive to poor CVs over 64 extracts (40 h batch analysis time, see above too). We did discuss this in the manuscript and showcased the PC and PE stability.

-The authors state "It is well known that diurnal changes of plasma lipidome, particularly of lipid mediators, readily occur". Please add one or two references to support the statement.

Answer: We included references.

-In the Discussion the authors state "...resolution of isobaric phospholipids with small differences in fatty acyl chains...". Maybe the authors want to revise this to say: "...resolution of isobaric phospholipids with major differences in fatty acyl chains...".

Answer: We did adjust accordingly. Thank you.

-In the Discussion the authors state "... detailed description of genetic regulation of extended...". Maybe the authors want to revise this to say: "... detailed description of genetic association of extended...".

Answer: We did adjust accordingly. Thank you.

-The authors state "... 4D lipidomics platform, highly recommend it for future translation..." Please use another word instead of 'recommend'.

Answer: We rephrased accordingly. Thank you.

-Figure 2bi and 2bii. Please show EIMs for 1:1 mixtures of PC 18:1_18:1 Cis and Trans and PG 18:1_18:1 Cis and Trans; and not EIMs for the analysis of the individual standards.

Answer: We did exchange the figure with 1:1 isomer mixtures. Thank you.

-Supplementary Figure 13b and 13d. Please specify in the legend whether the spectra have been smoothed or if it is raw TOF MS/MS data?

Answer: We did specify.

-Regarding PC O-24:2/20:4: This seems to be a false-positive identification. To the best of my knowledge, the 'key' fragment with measured m/z 485.3383 cannot be generated by the applied fragmentation mechanism (or at least show this occurs for a synthetic PC O- standard). This ion, with odd-nominal mass, would probably imply a radical-containing fragment; which is unlikely. Please show corresponding data from positive ion mode analysis (i.e., XIC, EIM, survey TOF MS spectrum and full MS/MS spectrum, including a zoom from m/z 350-650 (there should be detection of a fragment ion with m/z 526.3292, diagnostic for the neutral loss of the ether-link chain).

Answer: It is an interesting point. We included in the Figure zoomed m/z area to show the presence of ion with m/z 526.3292, hence the assignment is considered correct and supports also that the ion at m/z 485.3383 is generated from the structure PC O-24:2/20:4. Other PC O species, similarly show the same fragment type as the ion at m/z 485.3383. For reviewer information only, PC-O standards which we acquired in the context of another project during revision show a similar fragment ion type. Maybe the low abundance of this fragment type is indicative of its unlikely formation.

BTW: the calculated m/z for the acyl anion of 20:4 should be m/z 303.2330 (i.e., 303.232954) and not m/z 303.2323 (-2.2 ppm off). Similarly, the calculated m/z for the demethylated precursor should be m/z 860.6539 (i.e., 860.653864) and not m/z 860.6533 (-0.7 ppm off). It seems to me that there is a calculation mistake for especially fragment m/z values in the ClinLip Analyte List? It appears as if the authors have forgotten to account for the mass of the electron in their calculations?

Answer: Thank you for raising this issue. The m/z of the fragments in MS-Dial spectrum, which we used for this species curation in plasma were/are as given by MS-Dial for the in-silico fragmentation. Indeed, we didn't check if the electron mass was considered in the MS-Dial theoretical m/z of the ions. We included now also the theoretically calculated m/z of the fragments, with electron mass accounted for. The ClinLip Analyte List contains experimentally obtained fragmentation spectra and not in-silico fragmentation. As outlined in the manuscript, MS-Dial was used for curation of plasma lipid annotation and subsequent expansion of spectral library.

Reviewer #2 (Remarks to the Author):

No further comments.

Answer: Thank you.

Reviewer #3 (Remarks to the Author):

The authors generally addressed my comments well, and I appreciate the time and effort. Other than comments below their manuscript provides a methodology for semi-quantitative, high-coverage, reproducible, analysis of well annotated lipids for clinical studies. The proof will be in whether this method is widely adopted in the lipid community, which sadly often is not the case given most labs use separate methodologies. This is high-quality research with incremental improvement on lipidomic methodologies, using many common practices. My final decision is

that this should be published, but that it may not be revolutionary enough or novel enough for Nature Methods. I will leave that to the editor.

Answer: We thank the reviewer for the positive appreciation and remain hopeful that the lipidomic community will adopt this methodology and will further develop and leverage its potential in many applications.

Major:

While they clarify the purpose of semi-quantification for observing relative changes and biomarker discovery in the response document (which I agree fully with there use case), the manuscript still uses quantification in some places. Make sure to always state relative quantification, especially in the abstract. For example you mention in the abstract: "We present ... encompassing automated extraction, confident lipid annotation, and accurate quantification" so you are still misleading the reader... Relative quantitation, relative abundances, etc. should be used. Avoid concentration... etc.

Answer: We thank the reviewer for raising the issue of appropriate terminology. At the request of Reviewer 1, we used the term quantification as per the definition of the Lipidomics Standard Initiative; <https://lipidomicstandards.org/lipid-species-quantification>, and cited the link in the manuscript. We trust this correctly informs the reader.